# Long-read sequencing of an advanced cancer cohort resolves rearrangements, unravels haplotypes, and reveals methylation landscapes

## Graphical abstract

## Authors

Kieran O'Neill, Erin Pleasance, Jeremy Fan, ..., Marco A. Marra, Janessa Laskin, Steven J.M. Jones

## Correspondence

sjones@bcgsc.ca

## In brief

We present a resource dataset of 189 long-read-sequenced tumors from advanced cancer patients. Long-read sequencing allows detection of a range of features not easily detectable with more conventional short-read sequencing, including methylation, phasing, and complex rearrangements, with potential to enhance personalized cancer medicine.

## Highlights

- We present a rich data resource of 189 long-read-sequenced patient tumors

- Long-range phasing confirms biallelic tumor suppressor inactivation

- Complex SVs, viral integration, and ecDNA can be resolved

- Data phase promoter methylation in key cancer genes with therapeutic implications

O'Neill et al., 2024, Cell Genomics 4, 100674
November 13, 2024 © 2024 The Author(s). Published by Elsevier Inc.

CellPress

## Resource

# Long-read sequencing of an advanced cancer cohort resolves rearrangements, unravels haplotypes, and reveals methylation landscapes

Kieran O'Neill,[1,9,10] Erin Pleasance,[1,9] Jeremy Fan,[1,9] Vahid Akbari,[1,5,9] Glenn Chang,[1,9] Katherine Dixon,[1,9] Veronika Csizmok,[1,9] Signe MacLennan,[1,5,8,9] Vanessa Porter,[1,5,8] Andrew Galbraith,[1] Cameron J. Grisdale,[1] Luka Culibrk,[1] John H. Dupuis,[1] Richard Corbett,[1] James Hopkins,[1] Reanne Bowlby,[1] Pawan Pandoh,[1] Duane E. Smailus,[1] Dean Cheng,[1] Tina Wong,[1] Connor Frey,[2] Yaoqing Shen,[1] Eleanor Lewis,[1] Luis F. Paulin,[3] Fritz J. Sedlazeck,[3] Jessica M.T. Nelson,[1] Eric Chuah,[1] Karen L. Mungall,[1] Richard A. Moore,[1] Robin Coope,[1] Andrew J. Mungall,[1] Melissa K. McConechy,[1] Laura M. Williamson,[1] Kasmintan A. Schrader,[4,5] Stephen Yip,[6] Marco A. Marra,[1,5,8] Janessa Laskin,[7] and Steven J.M. Jones[1,5,*]

[1]Canada's Michael Smith Genome Sciences Centre at BC Cancer, Vancouver, BC, Canada
[2]Department of Medicine, University of British Columbia, Vancouver, BC, Canada
[3]Human Genome Sequencing Center, Baylor College of Medicine, Houston, TX, USA
[4]Hereditary Cancer Program, BC Cancer, Vancouver, BC, Canada
[5]Department of Medical Genetics, University of British Columbia, Vancouver, BC, Canada
[6]Department of Pathology and Laboratory Medicine, University of British Columbia, Vancouver, BC, Canada
[7]Department of Medical Oncology, BC Cancer, Vancouver, BC, Canada
[8]Michael Smith Laboratories, University of British Columbia, Vancouver, BC, Canada
[9]These authors contributed equally
[10]Lead contact
*Correspondence: sjones@bcgsc.ca

## SUMMARY

The Long-Read Personalized OncoGenomics (POG) dataset comprises a cohort of 189 patient tumors and 41 matched normal samples sequenced using the Oxford Nanopore Technologies PromethION platform. This dataset from the POG program and the Marathon of Hope Cancer Centres Network includes DNA and RNA short-read sequence data, analytics, and clinical information. We show the potential of long-read sequencing for resolving complex cancer-related structural variants, viral integrations, and extrachromosomal circular DNA. Long-range phasing facilitates the discovery of allelically differentially methylated regions (aDMRs) and allele-specific expression, including recurrent aDMRs in the cancer genes *RET* and *CDKN2A*. Germline promoter methylation in *MLH1* can be directly observed in Lynch syndrome. Promoter methylation in *BRCA1* and *RAD51C* is a likely driver behind homologous recombination deficiency where no coding driver mutation was found. This dataset demonstrates applications for long-read sequencing in precision medicine and is available as a resource for developing analytical approaches using this technology.

## INTRODUCTION

Cancer is a multifaceted, heterogeneous disease that arises from a diverse array of genetic alterations. Comprehensive profiling methods have emerged as fundamental tools for deciphering the distinct genetic landscape and biology of each tumor and identifying therapeutic vulnerabilities.[1,2] While panel-based sequencing approaches have become routine in clinical settings,[3,4] the significance of whole-genome and whole-transcriptome analysis (WGTA) has progressively gained recognition in both pediatric and adult cancers.[5–8] WGTA reveals driver mutations, gene fusions, expression alterations, and genome signatures, significantly contributing to our understanding of cancer genome landscapes and informing tailored therapeutic choices for patients.[9]

Cancer profiling has to date been predominantly reliant on short-read sequencing methods, which, while very successful, have inherent challenges and constraints due to read length.[10] More recently, long-read sequencing, exemplified by Pacific Biosciences and Oxford Nanopore Technologies (ONT), can routinely produce reads of tens of thousands of bases, which impacts complex structural variant (SV) calling and ultra-long variant phasing.[11] In contrast, phasing to identify which variants occur on the same chromosome on the basis of short reads alone requires parental genotypes or statistical inference from reference populations. Another notable feature of sequencing native DNA using long-read technologies is the simultaneous detection of 5-methylcytosine.[12,13] Short-read methodologies require separate samples with an experimentally intensive assay for methylation detection (for example, bisulfite sequencing).

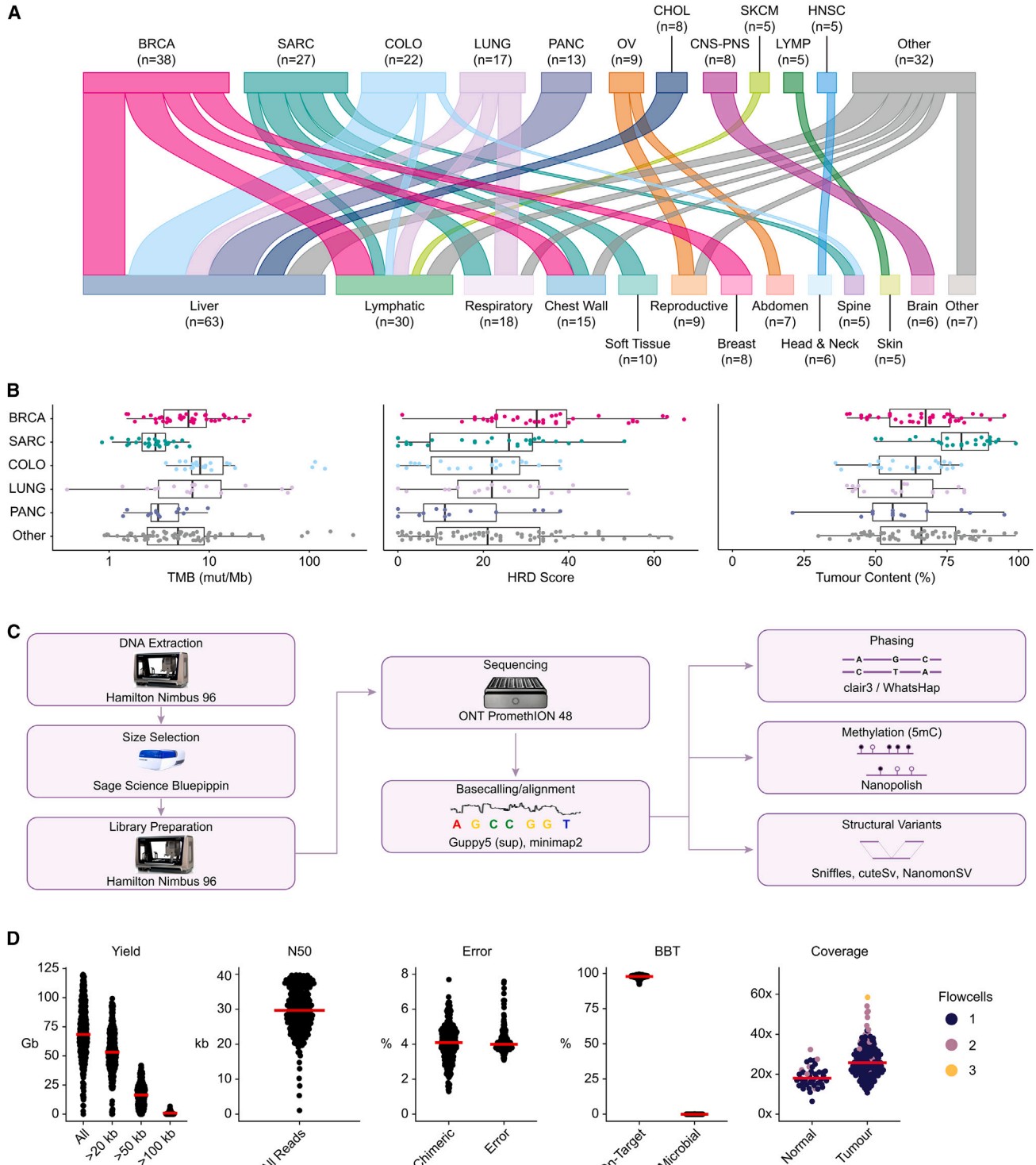

**Figure 1. Nanopore long-read sequencing of an advanced cancer cohort**

(A) Tumor types (top) and metastatic sites (bottom) for patient samples. Each patient is represented once; tissue groups with fewer than five samples are shown under "other."

(B) Genomic features of tumors by type. TMB, tumor mutation burden; mut/Mb, mutations per megabase; HRD, homologous recombination deficiency; BRCA, breast; SARC, sarcoma; COLO, colorectal; PANC, pancreatic. Boxplots represent the median and upper and lower quartiles of the distribution, and whiskers represent 1.5× interquartile range (IQR). "Other" tumor group includes all tumors not in the five most common tumor types, *n* = 72.

*(legend continued on next page)*

DNA methylation is a key driver of many cancers, and characterizing DNA methylation has potential diagnostic, prognostic, and therapeutic applications.[14,15] Explorations of long-read sequencing in small cohorts of adult and pediatric tumors have proven fruitful, unveiling complex rearrangements, viral integrations, and tumor-specific methylation alterations.[16–18]

To achieve the potential of long-read sequencing in cancer genomics, the development of a comprehensive suite of analytical methods tailored explicitly for tumor analysis is imperative. Existing tools are often unsuitable for cancer analysis or have been tested solely on cell-line data.[19,20] Patient-derived cancer samples encompass diverse features, including tumor heterogeneity, normal cell contamination from distinct tissues, and variability in mutation burden,[1,2] mandating analytical method refinement. To date, the absence of a sizable cohort of patient-derived cancer cases subjected to long-read sequencing has impeded progress in developing cancer-specific analytical approaches.

Here, we present data from the Long-Read Personalized OncoGenomics (POG) cohort of 189 tumor samples obtained from 181 patients enrolled in the POG program (NCT #02155621), sequenced using an ONT PromethION as part of the Marathon of Hope Cancer Centres Network. Each case in this cohort has also been studied using Illumina short-read normal whole-genome sequencing (WGS), tumor WGS, and tumor RNA sequencing (RNA-seq). Our analyses illustrate the broad potential utility of this dataset in personalized oncogenomics. All data have been deposited in the European Genomics Archive as a resource for developers of software for tumor characterization from nanopore long-read data.

## RESULTS

### The Long-Read POG cohort

Samples for long-read sequencing, previous short-read sequencing data, and short-read analysis were provided by the POG program, a precision medicine initiative that seeks to integrate WGTA into the clinical care of advanced cancer patients[2,6] (Table S1). The criteria used to select samples for long-read analysis included mutations in epigenetic modifiers, SV burden or homologous recombination deficiency (HRD), presence of human papillomavirus (HPV), and sufficient material and tumor content. This Long-Read POG cohort consists of 189 tumor samples from 181 patients. Of these, 43 tumor samples from 41 patients have matched normal nanopore sequencing allowing for somatic variant detection from long-read data. There were 26 cancer types represented, with the most common being breast (n = 38, 20%), sarcoma (n = 27, 14%), and colorectal (n = 22, 12%) (Figure 1A). The majority of the tumors (n = 144, 76%) were from biopsies of metastatic sites, frequently liver (n = 63, 33%) and lymph nodes (n = 30, 16%), while the rest were local recurrences or refractory disease (n = 45, 24%). Patients received between zero and nine lines of systemic therapy before sequencing and had a median overall survival of 34 months from

diagnosis of advanced disease and 17 months from biopsy (Figures S1A and S1B; Table S1). The tumor mutation burden (TMB) determined from short-read variant analysis varied from 0.4 to 274 mutations per megabase (mut/Mb, median: 4.9), with seven cases exhibiting microsatellite instability (MSI; Figure 1B). HRD, as measured by the HRDetect score on short-read data, was considered high for 26 samples (14%), the majority of which (14/26) were breast or ovarian cancers. The tumor content ranged from 21% to 100% (median 67%), with estimates of immune infiltration provided in Table S1.

Automated library construction and nanopore sequencing on the PromethION platform (Figures 1C, S1C, and S1D) yielded a median of 17.5- and 26-fold haploid genome coverage for normal and tumor samples, respectively (Figure 1D). The reads had a median N50 length of 31.3 kb, with the longest read spanning 1,036,455 bp. Reads longer than 20 kb accounted for 77.8% of the sequence data. Median base error (the edit distance of aligned reads to the GRCh38 reference) was 4%. Chimeric artifacts were present in a median of 4.1% of the reads. Assessment for microbial contamination showed fewer than 0.2% of reads matching microbial taxa in any sample, which is below the false discovery rate for the method used.[21]

### Nanopore sequencing reveals novel complex SVs

We sought to evaluate the potential of nanopore sequencing, combined with currently available software, for detecting SVs in cancer samples. To this end, we applied four variant callers, two with the ability to call somatic events and two without (Figures 2A and S2; Table S2). We began by compiling a list of well-established oncogenic fusion events that were previously identified using short-read sequencing in this cancer cohort.[22] Of these, 8/8 were successfully identified in the nanopore sequencing data using a combination of SV callers (Table S3), despite lower median sequencing depth of 26× for long reads compared to short reads, typically at 80×.

We further compared somatic long-read calls with high-confidence somatic calls previously made in short-read data[2] (see STAR Methods). Of those, 1,919 (54.1%) duplications, 1,943 (57.1%) inversions, 3,358 (37.6%) deletions, and 7 (<1.0%) insertions were consistent between the short- and the long-read datasets (see STAR Methods) (Figure 2B). To understand the disparities between the calls made on the different platforms, we manually reviewed those calls that overlapped cancer genes. The absence of calls in the nanopore results was attributed to lower coverage in 3/4 (75%) of samples. Conversely, all of the events unique to the nanopore calls (6/6) showed evidence in the underlying nanopore sequence alignment, but not in the short read (Table S4). In 4/6 samples, this difference was attributed to low-complexity regions, which could not be mapped with shorter reads. The remaining two calls were complex variants that could not be fully resolved by the short-read callers.

We examined these two complex SVs in depth. The first was a loss-of-function inversion, deletion, and duplication in *SMG1* in a colorectal adenocarcinoma sample (POG117; Figure 2C). The

(C) Schematic overview of the laboratory methods and primary analysis for this cohort.

(D) Fold coverage per sample sequenced and per-flow cell quality control statistics. Red bars indicate medians. Median yield of 68.4 Gbp per flow cell. Using BioBloom tools,[21] a median of 97.8% of reads matched the human reference, while no sample showed more than 0.2% of reads matching microbial taxa.

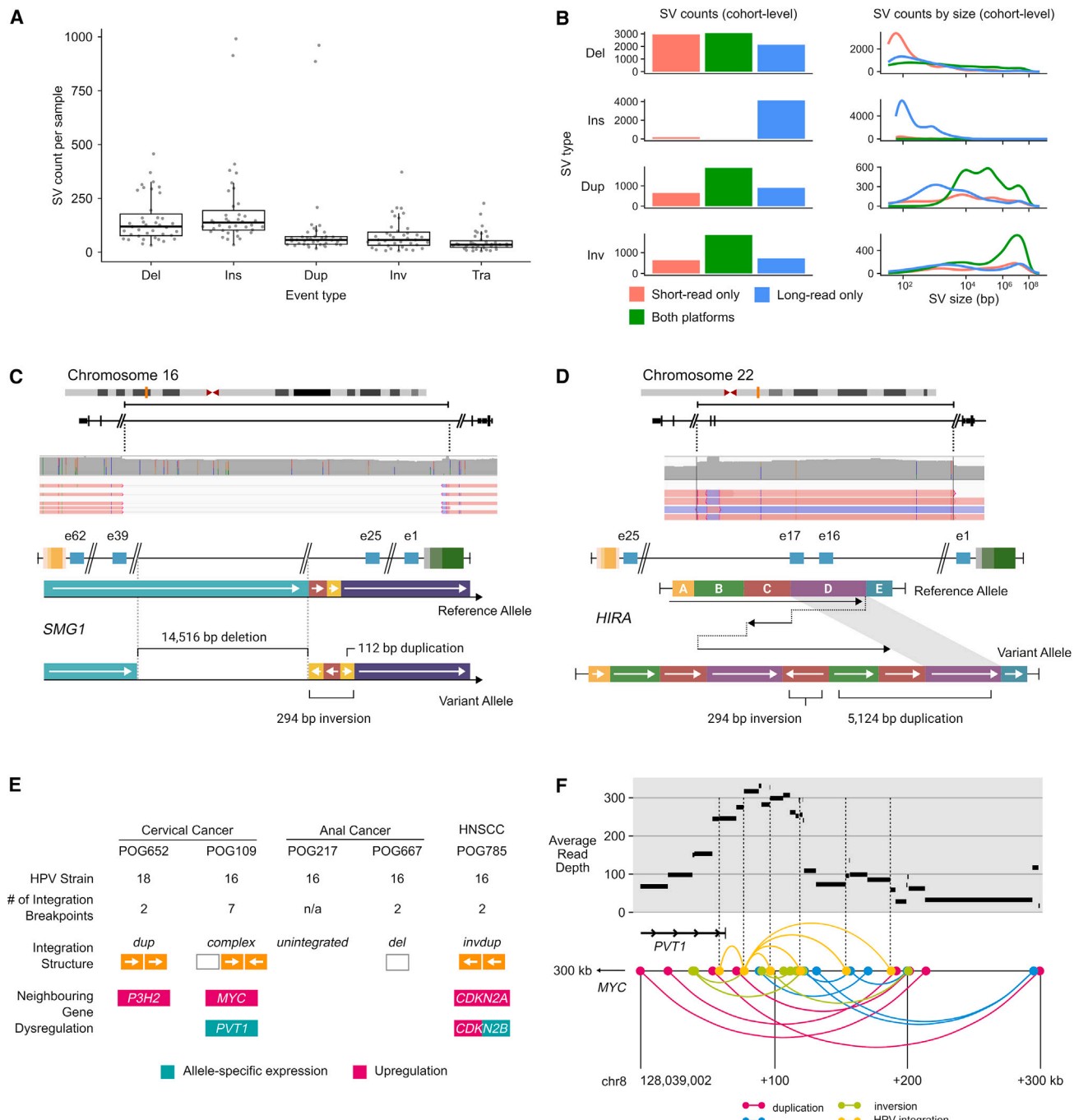

**Figure 2. Structural variants**

(A) Per-sample counts of somatic SV calls in samples with matched normal (*n* = 43). Boxplots represent median, upper, and lower quartiles; whiskers represent 1.5× IQR. Del, deletion; Ins, insertion; Dup, duplication; Inv, inversion; Tra, translocation.

(B) Concordance of SV calling between platforms, summed across cohort.

(C) Schematic of a resolved complex foldback inversion affecting *SMG1*, including a deletion of exons 26–38, detected only in the nanopore data.

(D) Schematic of a resolved complex foldback inversion affecting *HIRA*, including duplication of exons 16–17, detected only in the nanopore data.

(E) Features of HPV integration characterized using nanopore sequencing in the five tumors with HPV.

(F) Diagram of a complex rearrangement (bottom) and alterations in read depth (top) involving HPV integration sites in a cervical cancer (POG109).

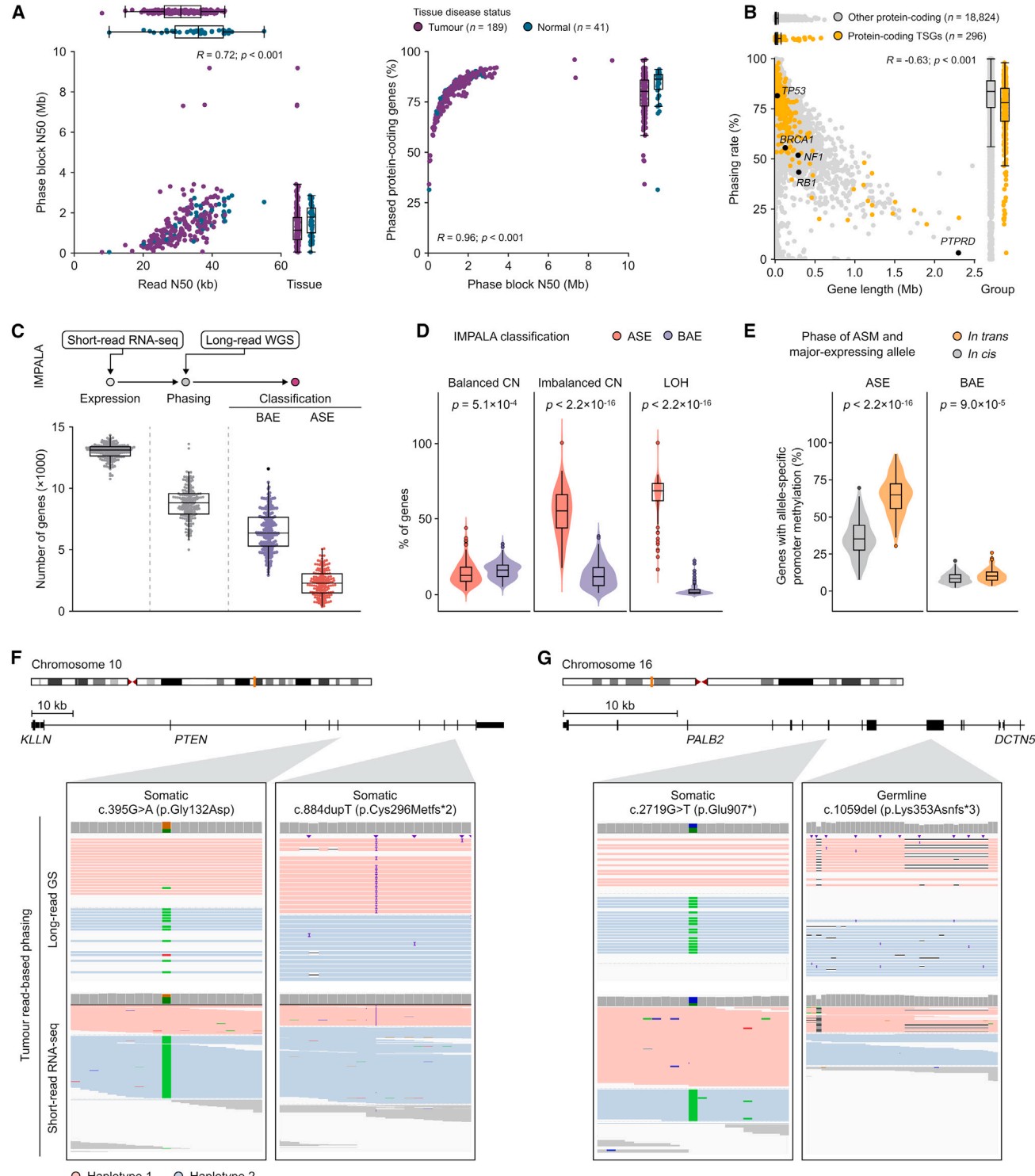

**Figure 3. Phasing**

(A) Spearman's correlation between read length, phase block size, and phasing rate for Ensembl 100 protein-coding genes (plus promoters) across normal and tumor tissues.

(B) Spearman's correlation between gene length and phasing rate for protein-coding genes (percentage of tumors in which a gene plus promoter could be fully phased).

second was an inversion with multiple duplication events that predicted a frameshift and likely loss of function of one allele of the tumor suppressor *HIRA* in a metastatic breast cancer sample (POG279; Figure 2D). These complex SVs showed an overlapping breakpoint in an L1MB4 long interspersed nuclear element (LINE) and an AluY element, respectively, underscoring the capacity of long-read sequencing to resolve some SVs in repetitive regions.

The greatest disparity in somatic calls was a notably greater number of insertions called in the nanopore data (Figure 2B). We manually reviewed the underlying sequence data for those affecting cancer genes. Of these, 8/14 were identified as miscalls resulting from difficult-to-map regions. The remaining 6/14 had underlying nanopore sequence evidence but were missed or mischaracterized by short-read callers (Table S5; Figure S3). Notably, all 14 of these insertion calls were made by a single SV caller, nanomonSV. This highlights the potential for long reads but also the need for further development of nanopore somatic SV calling software.

Four samples exhibited clear outlier behavior in terms of the number of SV calls of a particular type (Figure S4A). Two patients, POG884 and POG986, had 10-fold more insertion calls than the median, but only in the nanopore calls, accounting for 1,837/4,126 of the nanopore-only insertion calls mentioned above. These were the only two cases (of those with somatic calls) with MSI. On examining the insertions themselves, we found that the majority (71.5%) overlapped short tandem repeat regions, consistent with MSI repeat expansions (Figure S4B).[23] Two other cases, POG111 and POG147, exhibited high nanopore-unique inversion frequencies. These showed copy number alterations (CNAs) and inversion profiles that have been previously characterized as "tyfonas"[24] (Figure S5).

### Oncoviral integrations detected by long reads impact surrounding gene expression
HPV infection is the driving cause of cervical cancer and implicated in many head-and-neck and anogenital cancers. HPV integration into the host genome is frequent, and integration events often involve a complex combination of structural alterations and multiple copies of the 8 kb viral genome. This complexity makes mapping with short reads difficult. We investigated the ability of long-read sequencing to reconstruct HPV integration events and their effects. We investigated five samples with HPV previously detected in short-read data and confirmed in this study. Of these, four had HPV integration detected, each at a single host genome location (Figure 2E; Table S1). We identified three of these events as simple, meaning that they were made up of only two integration breakpoints. The remaining event, in POG109, was complex, incorporating seven HPV-to-host genome breakpoints within a 130 kb region in the 8q24 locus,

300 kb downstream (3′) of *MYC* and overlapping the *MYC*-associated long non-coding RNA (lncRNA) *PVT1* (Figure 2F). This event also overlapped several SVs and oscillating copy number states, which resembled focal chromothripsis (Figure S6A).

Three of the integration events were associated with elevated expression (≥85th percentile[2]; Figure S6B) of neighboring cancer genes, including *P3H2*, *MYC*, *CDKN2A*, and *CDKN2B*. POG109 exhibited allele-specific expression (ASE) of *PVT1*, the gene it overlapped, with higher expression from the haplotype containing the integration. POG785 (which had overexpression of *CDKN2A* and *CDKN2B*) exhibited ASE of *CDKN2B*, also with higher expression from the haplotype containing the integration.

### Long-range phasing enables resolution of double hits to tumor suppressors
We assessed the ability of nanopore sequencing to enable long-range phasing, particularly of tumor suppressor genes (TSGs). The biallelic inactivation of TSGs is an important mechanism of tumor formation, with potential biological and clinical significance for informing diagnosis, disease prognosis, and/or response to therapy. We found that phase block size was strongly correlated with read length (Spearman's rho 0.72, $p \leq 2.2 \times 10^{-16}$; Figure 3A), and the cases with the longest phase blocks also had high TMB (Figure S7). Phasing was able to completely resolve the haplotypes of the majority of genes in each sample, from promoter to 3′ end (median 85%, interquartile range [IQR] 79.1%–89.1%). This included most putative and known TSGs (Figure 3B), although longer genes were completely phased less often (Spearman's rho $-0.65$, $p \leq 2.2 \times 10^{-16}$). However, several notable tumor suppressors of modest size, including *BRCA1*, *NF1*, and *RB1*, could be phased completely in only around half of tumors, suggesting other locus-related features may reduce their phasing potential. Further investigation showed that this was due to lower density of phasing SNPs and greater density of repetitive or paralogous sequences that may not be fully resolved at modest read lengths (Figure S7). In tumor genomes, ploidy, genomic instability, loss of heterozygosity (LOH), and somatic variation may further influence global and local phasing.

Biallelic tumor suppressor inactivation may occur via different combinations of LOH and small mutations. When two small inactivating mutations are called by short-read sequencing in the same TSG, it is often assumed that they are in *trans*. In the Long-Read POG cohort, cases with two or more small somatic variants in TSGs with potential biological and/or clinical significance were identified by retrospective review of the genomic report issued at the time of POG analysis. There were 30 cases identified with double somatic variants in at least one TSG. Among 33 pairs of variants (with three variants in one case), 19 across 18 cases could be phased by long reads (Table 1).

---

(C) Summary of IMPALA results for the cohort, showing number of genes with sufficient expression to be considered (<1 TPM), number with sufficient expression and at least one phasing SNP, and their final classification as having allele-specific expression (ASE) or balanced allelic expression (BAE).

(D) Percentage of genes in regions of the tumor genome with balanced copy number (CN), imbalanced CN, or LOH that were classified as ASE or BAE.

(E) Percentage of genes with allele-specific promoter methylation by the relative phase of the major expressed allele for ASE and BAE genes. Boxplots represent median, upper, and lower quartiles; whiskers represent 1.5× IQR for (C), (D), and (E). The *p* values are Wilcoxon rank-sum test for (D) and (E).

(F and G) Examples of biallelic variants in tumor suppressor genes with ASE (F) and BAE (G). Reads are colored by predicted haplotype from long-read-based phasing, and reads that could not be assigned to a haplotype are colored in gray.

**Table 1. Phasing of tumor suppressor gene small variants**

| Phase | Origin | Case ID | Cancer | Gene | Allele A | Allele B |
|---|---|---|---|---|---|---|
| In *trans* | germline and somatic | POG1000 | PANC | *ATM* | exon 9 deletion (germline) | c.3577−14_3585delinsC (somatic) |
| In *trans* | germline and somatic | POG792 | CHOL | *BRCA2* | p.(Leu2092Profs*7) (germline) | p.(Val2385Phefs*9) (somatic) |
| In *trans* | germline and somatic | POG976 | CHOL | *PALB2* | p.(Lys353Asnfs*3) (germline) | p.(Glu907*) (somatic) |
| In *trans* | germline and somatic | POG604 | BRCA | *TP53* | p.(Arg213*) (germline) | p.(Glu180*) (somatic) |
| In *trans* | double somatic | POG295 | COLO | *APC* | p.(Cys417Valfs*37) | p.(Tyr1376Cysfs*9) |
| In *trans* | double somatic | POG720 | COLO | *APC* | p.(Ala591Profs*19) | p.(Pro1427Lysfs*44) |
| In *trans* | double somatic | POG777[a] | OV | *ARID1A* | p.(Gln1401*) | p.(Met1595Val) |
| In *trans* | double somatic | POG130 | COLO | *MLH1* | p.(Glu297*) | p.(Phe530Serfs*5) |
| In *trans* | double somatic | POG581 | SKCM | *NF1* | p.(Ser879*) | p.(Gln1188His) |
| In *trans* | double somatic | POG239 | SECR | *NOTCH1* | p.(Ala1349Leufs*53) | p.(Cys942Tyr) |
| In *trans* | double somatic | POG777[a] | OV | *PIK3R1* | p.(Thr371Ilefs*5) | p.(Lys459del) |
| In *trans* | double somatic | POG352 | UCEC | *PTEN* | p.(Arg130Gly) | c.1026+1G>T |
| In *trans* | double somatic | POG778 | BRCA | *PTEN* | p.(Tyr27Asp) | p.(Leu247Phefs*6) |
| In *trans* | double somatic | POG958 | BRCA | *PTEN* | p.(Ile32Asn) and p.(Ala34Gly) | c.490_492+1del |
| In *trans* | double somatic | POG884 | ESCA | *RB1* | p.(Arg255*) | p.(Met484Valfs*8) |
| In *trans* | double somatic | POG021 | LUNG | *TP53* | p.(Val272Leu) | p.(Gly154Val) |
| In *trans* | double somatic | POG680 | HNSC | *TP53* | p.(Pro82Leu) and p.(Ser127Pro) | p.(Arg282Trp) |
| In *cis* | double somatic | POG446 | BRCA | *KAT5* | p.(Ile37Met) and p.(Ser135Phe) | – |
| In *cis* | double somatic | POG507[b] | BRCA | *PTEN* | p.(Pro38Ser) and p.(Phe278Leu) | – |

[a]This case is represented twice as it includes two genes with hits on both alleles.
[b]Double somatic variants occurred in the context of copy loss of the other allele, consistent with biallelic events in *PTEN*.

Variants for which phase could not be confirmed included variants supported by only one read, variants not within a phase block, and variants supported by reads with conflicting haplotype assignments. The majority ($n$ = 17) of pairs were found to occur in *trans*, while two were found to occur in *cis*: double somatic variants in *PTEN* (POG507) and *KAT5* (POG446). Notably, the double somatic variants in *PTEN* were found to occur on the opposite allele from a heterozygous somatic *PTEN* deletion, suggesting an alternate mechanism of biallelic loss.

## Long-range variant phasing facilitates the detection of ASE and linkage to genomic events

ASE is an imbalance in expression between alleles of a gene, which is typically mediated by CNA or *cis*-acting regulatory mechanisms.[25,26] Long-range phasing offers the potential to more accurately determine ASE and link it to genomic lesions within the same or nearby genes.[16,27] We used the IMPALA pipeline to examine ASE in the Long-Read POG cohort.

We found ASE in an average of 26.5% of the phased genes within each sample (SD = 12.2%) (Figure 3C). CNAs have been identified as the primary drivers of ASE genes in cancer cells,[28,29] and our results recapitulated this. Within this cohort, ASE genes tended to overlap regions of LOH ($p \leq 2.2 \times 10^{-16}$) and copy number imbalance ($p \leq 2.2 \times 10^{-16}$), whereas genes with biallelic expression (BAE) overlapped copy-number-balanced regions (Wilcoxon rank sum $p$ = 0.02) (Figure 3D). We further noted a significant positive correlation between the CNA allelic ratio and the major expressed allele frequency (Pearson $r$ = 0.63, $p \leq 2.2 \times 10^{-16}$). Considering specifically nonsense mutations resulting in premature stop codons (Figure S8), in balanced copy

number regions these are found more frequently in genes that are ASE ($p = 1.5 \times 10^{-4}$) and are more likely to be on the minor expressing allele ($p = 1.6 \times 10^{-13}$); this is consistent with loss of expression of the mutated allele due to nonsense-mediated mRNA decay. ASE may also be due to epigenomic dysregulation. Examining ASE in regions with balanced copy number, we found a significantly greater proportion of genes with promoter allelic methylation (mean = 0.16) than with BAE (mean = 0.01) (Wilcoxon rank sum, $p = 3.5 \times 10^{-23}$). Moreover, allelic promoter methylation was more commonly found in *trans* with the major expressed allele ($p \leq 2.2 \times 10^{-16}$) (Figure 3E), indicating lower expression of the methylated allele.

ASE can be used to validate the downstream expression effects of aberrant *cis*-acting regulatory mechanisms and further ascertain biallelic TSG inactivation. For example, *PTEN* in POG041 shows ASE with major expressed allele frequency of 0.67. A frameshift and a missense mutation (rs121909241) were found on the minor and major expressed allele, respectively, which represents a double-hit knockout scenario and ASE consistent with larger impact on expression from the truncating mutation (Figure 3F). POG976, with both *PALB2* somatic and germline variants, confirmed by phasing to be opposite alleles, shows BAE, as both alleles are impacted by truncating events (Figure 3G). Consistent with loss of function of *PALB2*, this cholangiocarcinoma was characterized by strong mutational signatures of HRD (Table S1).

The most frequent ASE gene in this cohort was *DUSP22*, in 122/135 samples in which it was expressed and could be phased, with a median major expressed allele frequency of 0.95. *DUSP22* expression is associated with poorer survival in

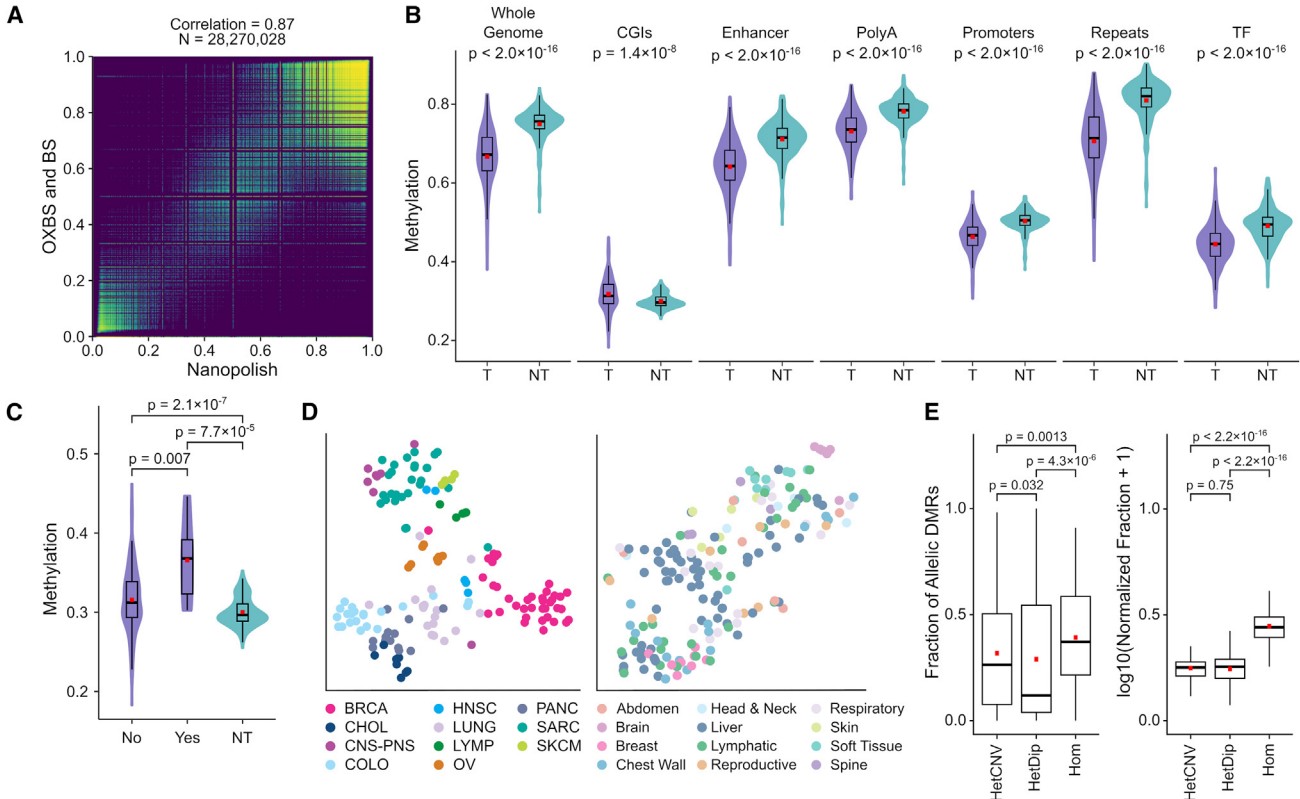

**Figure 4. Methylation**

(A) Correlation of nanopolish methylation frequency with WGBS for POG044. BS, bisulfite sequencing; OXBS, oxidative bisulfite sequencing.

(B) Average methylation across tumors (T) compared with public WGBS methylation data from normal tissues and cells (NT), genome wide and at different genomic regions.

(C) Average methylation at CGIs in POG cases with either IDH-activating or TET-inactivating mutations (yes) compared with the remainder of the cohort (no) and public normal tissue (NT).

(D) tSNE plots based on DNA methylation at regulatory regions, compared with tumor type (left) and biopsy site (right).

(E) aDMR distributions by copy number (CN). Heterozygous diploid (HetDip) indicates CN-balanced regions. Heterozygous copy number variant (HetCNV) indicates CNV regions with both parental alleles. Homozygous (Hom) indicates LOH. All $p$ values are Wilcoxon rank-sum test. Boxplots represent median, upper, and lower quartiles; whiskers represent 1.5× IQR.

low-grade lymphomas.[30,31] It also shows tissue-specific imprinting during brain development.[32,33] A survey of normal tissues from the Genotype-Tissue Expression (GTEx) project showed that only 9.12% (542/5940) of *DUSP22*-expressing samples showed ASE.[34] In the Long-Read POG cohort, 63.11% of tumor samples with ASE in *DUSP22* showed allele-specific promoter methylation in *trans* and 81.15% showed allele-specific gene body methylation in *cis* with the major expressed allele (Figure S9). Although allele-specific promoter methylation is found in both normal and tumor samples, allelic loss of gene body methylation is a tumor-specific phenomenon ($p \leq 2.2 \times 10^{-16}$). This suggests that reactivation of tissue-specific imprinting of *DUSP22* may play a role in tumorigenesis.

### DNA methylation derived from nanopore sequencing can reveal global methylation patterns and reflect tissue of origin

We evaluated nanopore-derived DNA methylation calls within this study for their potential in personalized oncogenomics. A comparison of DNA methylation detected using nanopore sequencing with whole-genome bisulfite sequencing (WGBS) calls from the same sample showed good correlation ($R = 0.87$, Figure 4A). In the Long-Read POG cohort, tumors displayed significant hypomethylation compared to normal WGBS data from best-match tissue types (Figure 4B; see STAR Methods). This hypomethylation was most distinct in repetitive regions. Notably, these regions are more readily mappable with long-read alignment.[35] The only genomic regions with hypermethylation in tumor samples compared to normal WGBS data ($p = 1.4 \times 10^{-8}$) were CpG islands (CGIs). These results are consistent with the previously described pattern of overall hypomethylation but focal hypermethylation in tumor DNA.[36]

TET enzymes are involved in active DNA demethylation and use α-ketoglutarate as a cofactor, which is a product of IDH enzyme activity. TET and IDH mutations are recurrent in cancer.[37] Loss of TET demethylase activity due to loss-of-function mutations can result in hypermethylation of tumor genomes. TET can also be inhibited by accumulation of metabolites

due to *IDH1* R132 and other gain-of-function mutations, also resulting in tumor hypermethylation.[37] Within the cohort, 10/189 samples (8/181 cases) had either *IDH1* R132 C/H or *TET1/2/3* candidate inactivating mutations detected using short-read sequencing (Table S1). Compared with other cases and normal tissue, cases with these mutations show similar methylation patterns at all regulatory sites except for CGIs (Figures 4C and S10). At CGIs, mutated samples show slight hypermethylation compared to the rest of the tumor samples and WGBS normal samples. TET enzymes demonstrate sequence specificity toward CGIs.[37–39] The slightly higher methylation only at CGIs in the mutated samples in our cohort supports the sequence specificity of TET enzymes and concurs with previous findings suggesting that the genome-wide hypomethylation in tumor samples is largely due to the passive DNA demethylation pathway.[40–42]

Methylation patterns can distinguish tissue and tumor type.[43,44] We performed t-distributed stochastic neighbor embedding (tSNE) analysis of methylation in this cohort as a coarse assessment of this. We observed that samples tended to group by tumor tissue of origin, irrespective of metastatic biopsy site (Figure 4D). This finding suggests the potential utility of nanopore-derived DNA methylation for detecting or confirming tissue of origin in advanced and metastatic cancers, as an adjunct or complementary analysis to the RNA approaches currently in use.[45]

### Patterns of allele-specific methylation in promoters and gene bodies are uncovered by long-range phasing

As shown earlier in this study and elsewhere, long reads enable long-range phasing, which extends to methylation information.[46] In this cohort, an average of 84% of the CpG sites within each sample could be phased (median = 86%; SD = 7.1%). We define the term "allelically differentially methylated regions" (aDMRs) to refer to genomic regions in which clusters of CpG sites display differential methylation between alleles. We detected 4.46 million (mean = 23.61K, SD = 14.64K) aDMRs across all tumor samples, with around 5-fold more in tumor samples than in their matched normals (mean = 4.7K, SD = 876; Figure S11A). The majority (79%) of the aDMRs mapped to heterozygous copy number variant (CNV) and LOH regions (Figure 4E). The number of aDMRs in each sample was positively correlated with the fraction of the genome in LOH regions (Pearson $r$ = 0.469, $p$ = 2.6 × $10^{-11}$) and negatively correlated with the fraction of the genome in heterozygous diploid regions (Pearson $r$ = −0.38, $p$ = 1.4 × $10^{-7}$).

We examined aDMRs in heterozygous diploid regions as potential sources of cancer-specific epigenetic dysregulation. We excluded aDMRs associated with normal cell function (linked to imprinting or random allele-specific methylation; see STAR Methods), leaving ~462K tumor-specific aDMRs. Most (76% of aDMRs) mapped to CGI, transcription factor (TF), promoter, enhancer, or poly(A) sites (Figure S11B). We detected 2,854 genes (8,511 transcripts) with recurrent aDMRs at their promoters, including the cancer genes *RET* and *CDKN2A*. TF enrichment analysis of these genes demonstrated significant enrichment for PRC1 and PRC2 complex protein subunits (adjusted $p$ < 0.05 from Enrichr; Figure S11C). PRC1 and

PRC2 are transcriptional repressive complexes involved in the regulation of developmental genes. Allele-specific methylation of their target genes in different tissues as well as hypermethylation of those genes in cancers have been reported.[47–49] This result suggests the preferential occurrence of aDMRs at genes likely involved in the stem cell-like properties of cancer.

We further examined the aDMRs affecting *RET* and *CDKN2A*. *RET* is a receptor tyrosine kinase and a well-known oncogene.[50] The *RET* aDMR overlapped an intragenic CGI alternate promoter (per Ensembl 100 transcript models) in 26 of the cohort samples (Figure 5A). Another 66 tumor samples showed methylation (>25% average methylation) at this promoter. Those cases with intragenic promoter methylation (allelic or non-allelic) had significantly greater overall *RET* gene expression than those without methylation ($p$ = 0.0071) and normal GTEx tissues ($p$ = 2.2 × $10^{-6}$; Figure 5B).

*CDKN2A* is a well-known TSG that has a series of intragenic CGI promoters, which showed recurrent aDMRs in nine samples (Figure 5C) and methylation (>25% average methylation) in 140 samples. As for *RET*, cases with methylation (allelic or non-allelic) at this intragenic promoter had significantly greater expression of *CDKN2A* compared to those without methylation ($p$ = 6 × $10^{-9}$) and normal GTEx data ($p$ < 2.22 × $10^{-16}$; Figures 5D and S12A). Despite *CDKN2A* being a tumor suppressor, it is sometimes upregulated in cancers, with associated increase in immune cell infiltration[51] (Figure S12B), suggesting that it may function as an oncogene. In both *RET* and *CDKN2A*, intragenic promoter methylation may be a novel means of inducing overexpression of these genes as part of tumorigenesis.

We further examined whether known promoter mutations are associated with ASE and methylation. We examined the most frequent hotspot non-coding mutations described in the POG570 cohort,[2] which are associated with the genes *TERT*, *PLEKHS1*, *ADGRG6*, and *AP2A1* (Table S6). As these mutations are still rare, in the Long-Read POG cohort we found only four or five mutations associated with the described hotspot regions in each of *PLEKHS1*, *ADGRG6*, and *AP2A1*, and with such few samples, we were unable to find associations with ASE or aDMRs. *TERT* promoter mutations were more common, found in a total of 13 samples. *TERT*, an oncogene encoding telomerase associated with maintaining telomeres in rapidly dividing cancer cells, is frequently overexpressed in tumors.[52] Mutations in the Long-Read POG cohort were indeed associated with higher *TERT* gene expression ($p$ = 0.0052; Figure 5E). As a TF, *TERT* generally has low transcripts per million (TPM), which makes measurement of ASE more challenging. However, examining the allele-specific data, we found more ASE for the allele containing the mutations (Fisher's exact test, $p$ = 0.006), in part due to the higher expression bringing more samples above the TPM ≥ 1 threshold used for calculating ASE (see STAR Methods). We also found no cases where *TERT* had statistically significant BAE. The *TERT* promoter is normally unmethylated,[52] which is also observed in Long-Read POG normal blood samples (Figure S12C). We found that *TERT* promoter mutations frequently overlapped tumor aDMRs but, interestingly, tended to be found on the less methylated allele (Wilcoxon rank-sum test, $p$ = 0.13 for average allele-specific methylation in 12

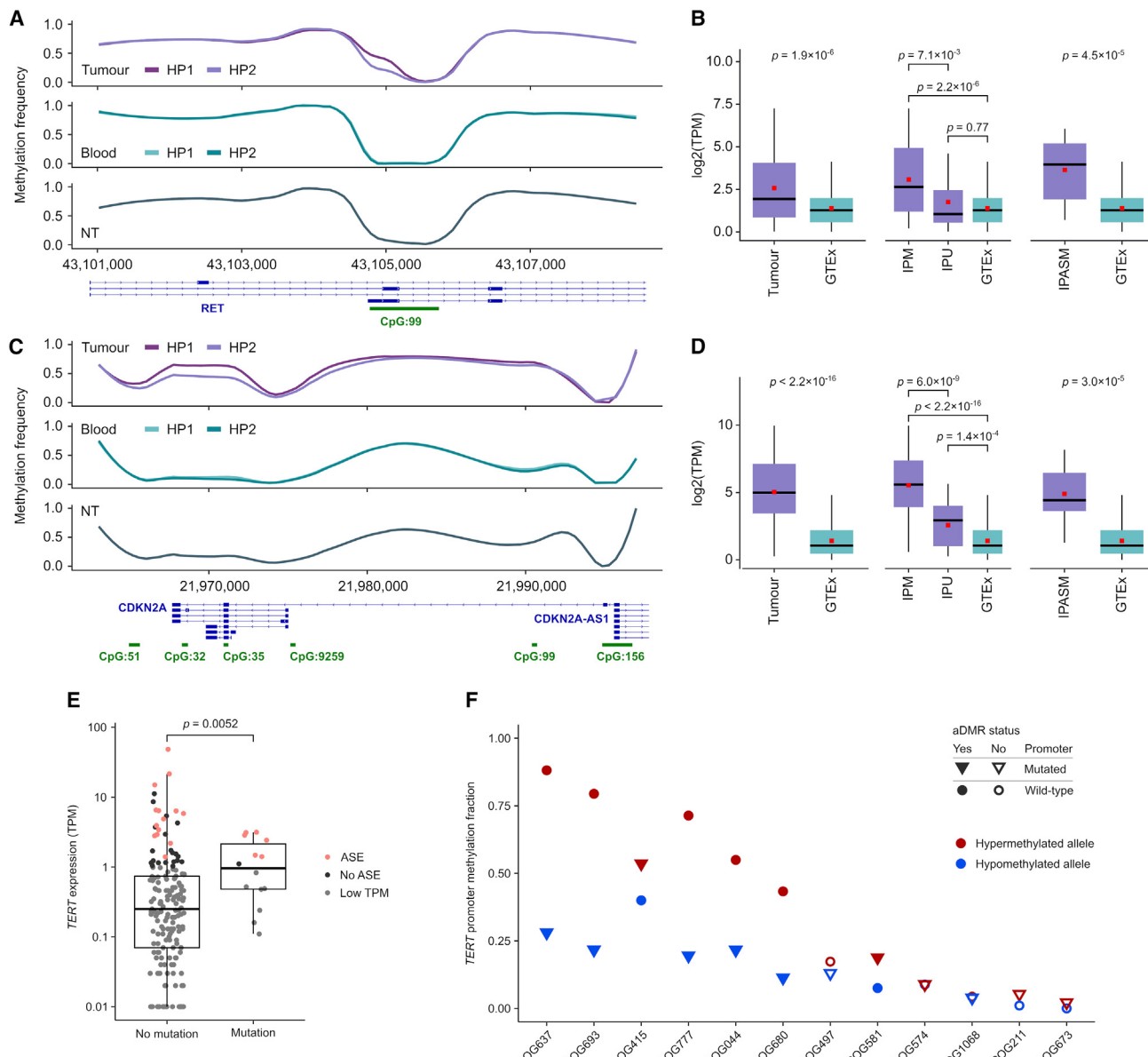

**Figure 5. Methylation in specific cancer genes**
(A) DNA methylation at *RET* intragenic promoter CpGs, compared with patient blood and normal tissue (NT).
(B) *RET* gene expression compared with GTEx normal tissues in (left) the whole cohort, (center) samples with >25% intragenic promoter methylation (IPM) vs. other samples (intragenic promoter unmethylated [IPU]), and (right) only those samples with an aDMR at the intragenic promoter (IPASM). Only samples with TPM > 1 were used for expression comparison.
(C and D) The same analysis as in (A) and (B) but for *CDKN2A*. Note that in (A) and (C) the haplotags were swapped so that HP1 represents the hypermethylated allele.
(E) TERT expression in samples with and without TERT promoter hotspot mutation at chr5:1,295,113 or chr5:1,295,135.
(F) Average allele-specific methylation of the core TERT promoter (153 CpGs in chr5:1,294,414–1,295,655). aDMRs are noted when an aDMR overlapping the hotspot mutation coordinates was identified by the software DSS and average allele-specific methylation differed by at least 0.1 between alleles within the defined core TERT promoter. All *p* values are Wilcoxon rank-sum test. Boxplots represent median, upper, and lower quartiles; whiskers represent 1.5× IQR. TPM, transcripts per million.

mutated cases; Figure 5F). In tumor cells, the *TERT* promoter is reported to be frequently methylated, but there has been some debate about whether *TERT* mutations and promoter methylation are mutually exclusive, and this has been explored in cell lines.[53] Our allele-specific data show that *TERT* promoter mutation and promoter methylation may indeed co-occur in a single patient tumor, but that these two alterations may be on different alleles.

**Table 2. Promoter methylation of HR genes in cases with HRD**

| POG ID | Tumor type | Analysis cohort | Gene | Fraction of methylated sites | LOH status | HRDetect |
|--------|-----------|-----------------|------|------------------------------|-----------|----------|
| POG425 | BRCA | breast and ovarian | *BRCA1* | 0.922 | LOH | 0.998 |
| POG507 | BRCA | breast and ovarian | *BRCA1* | 0.951 | LOH | 0.999 |
| POG804 | BRCA | breast and ovarian | *BRCA1* | 0.961 | LOH | 0.999 |
| POG356 | BRCA | breast and ovarian | *RAD51C* | 0.739 | LOH | 0.992 |
| POG846 | OV | breast and ovarian | *RAD51C* | 0.554 | LOH | 0.998 |
| POG894 | OV | breast and ovarian | *RAD51C* | 0.587 | LOH | 0.999 |
| POG277 | LUNG | other | *BRCA1* | 0.804 | HET | 0.007 |
| POG1041 | MISC | other | *BRCA1* | 0.765 | LOH | 0.319 |
| POG650 | HNSC | other | *BRCA1* | 0.657 | LOH | 0.635 |
| POG785 | HNSC | other | *RAD51C* | 0.576 | HET | 0.032 |
| POG266 | CHOL | other | *RAD51C* | 0.554 | HET | 0.103 |
| POG044 | CNS-PNS | other | *RAD51C* | 0.552 | HET | 0.001 |

### Epigenetic inactivation of DNA repair genes

HRD is especially prevalent in breast and ovarian cancers,[54] and its presence is relevant for therapeutic selection. HRD can arise due to inactivation of DNA repair genes by a combination of mutations and promoter methylation. We evaluated the promoter methylation frequencies of 51 homologous recombination (HR) genes in a combined breast (*n* = 40) and ovarian (*n* = 8) cohort (see STAR Methods). Three breast cancer samples showed *BRCA1* promoter hypermethylation (Table 2). *RAD51C* promoter hypermethylation was observed in one breast and two ovarian cancer samples (Table 2). All six samples exhibited high HRDetect scores (≥0.7; see STAR Methods) consistent with an HRD phenotype (Figure 6A). No deleterious somatic or pathogenic germline mutations were found in five HR genes (*BRCA1*, *BRCA2*, *ATM*, *PALB2*, and *RAD51C*) in these samples, suggesting that silencing of *BRCA1* and *RAD51C* was likely the primary cause of the observed HRD (Figure 6A).

Compared to the rest of the cohort, samples with *BRCA1* or *RAD51C* promoter hypermethylation showed reduced gene expression. This is consistent with methylation-induced transcriptional silencing (Figure 6B). Haplotype-specific methylation data revealed that the methylation was confined to a single allele in the HRD tumors with *BRCA1* (Figure 6C) and *RAD51C* (Figure 6D) promoter hypermethylation. Together with the observed LOH of the other allele, this is consistent with biallelic inactivation of these genes. Matched blood was available for all three *BRCA1* and one of the *RAD51C* cases. In all instances, the promoter alleles were unmethylated in the blood, suggesting that the tumor promoter methylation was a somatic event (Figures 6C and 6D).

HRD is observed in other cancers, albeit at lower frequencies. However, *BRCA1* and *RAD51C* promoter hypermethylation have been reported only in sporadic breast and ovarian cancer cases.[55,56] In this cohort, we identified promoter hypermethylation of *BRCA1* and *RAD51C* in a further three cases each (Table 2). Two of the samples with *BRCA1* promoter hypermethylation showed LOH of *BRCA1* and exhibited moderate HRDetect scores, suggesting potential HRD. In the remaining four samples, the other allele was intact and unmethylated, and the HRDetect score was low (Table 2), likely due to incomplete inactivation of the gene (Figure S13).

Outside of Lynch syndrome, inactivation of mismatch repair (MMR) genes and consequent MSI arise in sporadic cancers, usually from a combination of somatic MMR gene mutations or somatic hypermethylation at the promoter region. However, constitutional epimutations that result in germline promoter hypermethylation have been reported.[57] Our cohort included a patient, POG986, with lung squamous cell carcinoma and with multiple other previous primary cancer diagnoses, suggestive of Lynch syndrome. Previous clinical hereditary cancer multigene panel sequencing was uninformative, but targeted methylation testing of blood performed after short-read POG analysis showed constitutional methylation of *MLH1*, confirming Lynch syndrome. Long-read sequencing data from this study further confirmed monoallelic hypermethylation in the blood with no causative sequence variants (Figure 6E). In the tumor, somatic LOH of *MLH1* resulted in the loss of the wild-type *MLH1* allele, with hypermethylation on the remaining tumor allele (Figures 6E and 6F, POG986). Another tumor, endometrial cancer in patient POG041, showed hypermethylation on both tumor alleles of *MLH1* (Figure 6F). MMR deficiency was confirmed by immunohistochemical testing. No matched blood sample was long-read sequenced, but the absence of a germline mutation and the presence of biallelic methylation in the tumor suggest that the *MLH1* promoter methylation was a somatic event in this case.

### Genomic and epigenomic architecture of extrachromosomal DNA

We hypothesized that the added methylation and phasing information obtained from long reads would enable the reconstruction of both the genomic and the epigenomic structure of extrachromosomal DNA (ecDNA). Using AmpliconArchitect[58] on the short-read WGS data, we predicted the presence of 76 ecDNAs in 42/189 (22.2%) samples (Figure S14A; Table S1). A total of 1,283 genes were detected on the ecDNAs in our cohort, with 262 (20.4%) occurring on more than one ecDNA (Table S7). Importantly, 97 of these genes were oncogenes, 33 (34.0%) of which were recurrent in our cohort. *ZNF703* recurred most frequently, being present in ecDNAs in five samples. The presence of at least one ecDNA correlated with an increased

**CellPress**

**Cell Genomics**
Resource

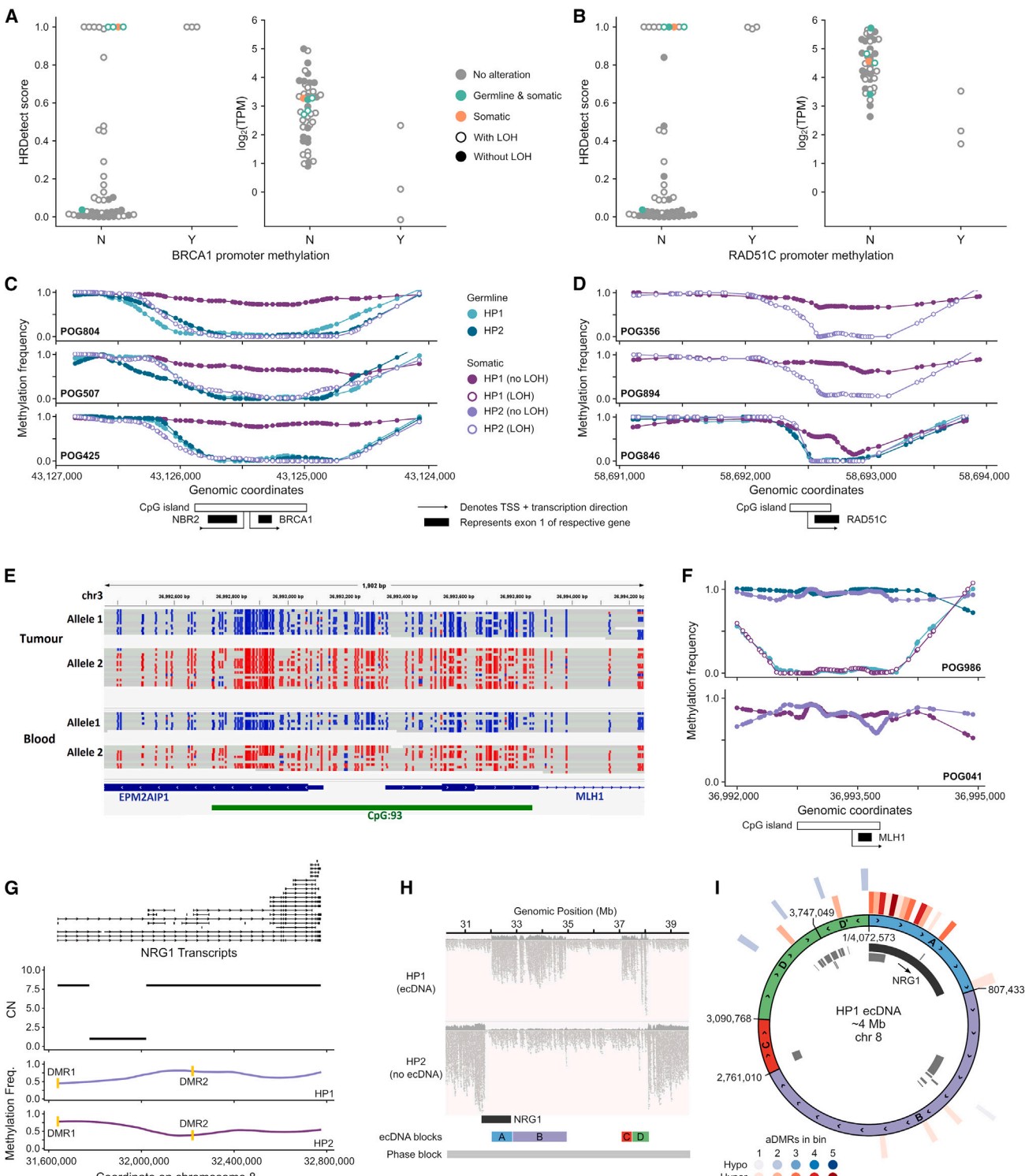

**Figure 6. Integrative analyses**

(A–D) *BRCA1* (A) and *RAD51C* (B) HRDetect scores (left) and expression values (right) for breast and ovarian samples with or without promoter methylation in *BRCA1* or *RAD51C*. Samples with deleterious alterations in five key HR genes (*BRCA1*, *BRCA2*, *ATM*, *PALB2*, and *RAD51C*) are colored orange (somatic) and green (germline and somatic). Haplotype-specific DNA methylation frequencies at the *BRCA1/NBR2* (C) and *RAD51C* (D) promoter regions in HRD samples (HRDetect score ≥ 0.7) with promoter methylation. Germline refers to a matched blood sample from the same individual.

(E) Haplotype-specific DNA methylation at the *MLH1* promoter in a lung squamous cell carcinoma sample with *MLH1* germline epimutation.

*(legend continued on next page)*

genomic complexity score (two-sided Student's t test; adjusted $p = 5.17 \times 10^{-7}$) and a trend toward an increased HRD score (two-sided Student's t test; adjusted $p = 0.0548$), which may be partially explained by the increased representation of breast cancers in the ecDNA+ cohort (Figure S14B).

We examined allelic methylation patterns within promoter regions for genes predicted to reside on ecDNAs, as a potential mechanism of ecDNA-specific dysregulation. We identified one breast cancer case with multiple aDMRs within an ∼4 Mb ecDNA, surrounding *NRG1*. Two aDMRs overlapped promoter regions for separate isoforms of the gene (Figure 6G). *NRG1* is a known cancer gene in breast cancer.[59–62] We were able to validate 4/5 of the breakpoints predicted by AmpliconArchitect by manual review of the long-read data (see STAR Methods). We were further able to assign the ecDNA to a single haplotype based on the nanopore long-range phasing and to confirm that the entire event fell within one phase block (Figure 6H; see STAR Methods). Both aDMRs in *NRG1* promoter regions showed hypermethylation of the amplified (ecDNA) allele. Furthermore, across the entire ecDNA region, the density of aDMRs was significantly greater within *NRG1* than in any other region of the ecDNA (Figure 6I). Although *NRG1* was expressed only from HP1, and was focally amplified in the genome, it was not overexpressed relative to other breast cancer samples within the cohort (Figure S14C). This finding suggests a regulatory mechanism by which an ecDNA-mediated amplification may be countered by promoter methylation of the amplified gene.

## DISCUSSION

We present the Long-Read POG cohort, a set of 189 tumor samples of diverse tumor types sequenced via the ONT sequencing platform. As the samples in this cohort have accompanying short-read DNA and RNA sequence data and associated clinical information, our study offers potential for advancing the understanding of SVs, viral integration, DNA methylation, and allelic information pertinent to cancer pathogenesis and diagnosis.

Long-read sequencing data enabled software-based calling of known oncogenic fusions and showed reasonable concordance with short-read SV calls genome wide. When examined, discordance was largely due to the much lower coverage of long reads or the inability to map breakpoints in the short reads. Long-read data enabled reconstruction of complex SV events that were undetected or mischaracterized in short-read data. Furthermore, long-read data enabled the direct detection of MSI expansions as SVs,[22,23] as well as the detection of "tyfona" signatures in sarcoma and melanoma as previously described.[24] Complex patterns of HPV integration could also be deconvoluted, along with their impact on surrounding cancer gene expression, as has been recently explored in cervical cancer.[63,64] Currently,

there is limited availability of somatic SV callers for the nanopore platform, but with the aid of datasets such as the Long-Read POG cohort, development of additional calling approaches will allow for improvement in this area. Our initial results unraveling the biology of a putative ecDNA in POG816 using long-read sequencing have been promising, suggesting future research could include a more comprehensive exploration of the remaining ecDNAs identified in short-read sequencing in this cohort.

Long-range phasing of variants using long reads facilitates the coverage of the vast majority of genes within single haplotype blocks. This enables phasing of double hits in tumor suppressors to ascertain biallelic loss of function. Moreover, when combined with short-read RNA data, phasing can link deleterious variants with ASE, providing further confirmation of multiple hits to tumor suppressors. We observed widespread recurrent ASE across tumors, as exemplified by the widespread ASE of *DUSP22*. The re-expression in tumors of genes that are typically developmentally restricted, such as *DUSP22*, has been suggested to be a potential source of neoantigen therapeutic targets.[65]

The ability of nanopore sequencing to provide methylation data within standard WGS without additional sequencing or bisulfite conversion is a significant advantage. We showed that methylation is associated with tissue of origin, suggesting the potential for tumor type classification.[66] Long-range phasing of methylation facilitates the exploration of aDMRs. Although we confirmed that the majority of aDMRs are driven by CNVs, we also observed epigenetic dysregulation in copy-neutral regions. Recurrent sites of aDMRs showed an enrichment for PRC1 and two binding sites, congruent with their well-known role in stem cell regulation and cancer.[47–49] In rare cases, this can be due to germline altered methylation, exemplified by the identification of germline inactivating *MLH1* methylation in a patient with clinically confirmed Lynch syndrome. This phenomenon has been described in both familial and sporadic Lynch syndrome patients, along with acquired hypermethylation of *MLH1* in endometrial cancers.[67] Long-read sequencing could be applied to identify causative epigenomic alterations in Lynch and other syndromes. Examination of the tumor microenvironment can be accomplished by deconvolving cell types from RNA-seq data, and we have included CIBERSORT results created from short-read RNA-seq for this cohort. Similar methods can be applied to methylation (5-mC) derived from long-read tumor sequence data, such as MethylCIBERSORT.[68]

We observed recurrent aDMRs in the intragenic promoters of *RET* and *CDKN2A*, with methylation being associated with increased expression of the canonical transcript. We further observed ASE-associated gene body methylation of *DUSP2*. The effect of intragenic promoter methylation on transcription is complex and bidirectional.[69–71] Gene bodies frequently

(F) Haplotype-specific DNA methylation frequencies at the *MLH1* promoter region in a lung squamous cell carcinoma sample with *MLH1* germline epimutation (top) and in a uterine endometrioid carcinoma sample with somatic *MLH1* promoter methylation (bottom).

(G) Haplotype-specific methylation and copy number for *NRG1* in breast cancer sample POG816. The 3′ amplification was included within an ecDNA. Promoter aDMRs are highlighted in yellow.

(H) Haplotype-phased long reads mapped to the ecDNA region.

(I) Circos plot of the *NRG1*-containing ecDNA, highlighting DMRs and methylation states. Inner track: gene annotations, with *NRG1* highlighted. Outer tracks: binned counts of aDMRs, showing substantial enrichment at the 5′ end of *NRG1*.

become methylated during active transcription,[69,70] which may silence intragenic promoters via transcription interference.[69–71] Conversely, intragenic promoter methylation can increase canonical transcript expression by reducing competition for RNA polymerase (Pol) binding or through regulating transcription elongation,[69] with evidence that this methylation regulates some oncogenes in cancer.[72,73] Further analysis of intragenic promoter methylation in *RET* and *CDKN2A* is needed to determine whether this change is a mere consequence of active transcription or a key regulator of expression. A wider survey of genes showing allele-specific gene body methylation either in this cohort or in others may also help to elucidate this phenomenon. Promoter methylation in the oncogene *NRG1* in one case was also notable for being ecDNA specific, a phenomenon that has not been well characterized to date.[74]

Our study expounds the significant role of gene promoter hypermethylation in HRD tumors, which often lack a clear underlying causative mutation.[75,76] Notably, the presence of *BRCA1* and *RAD51C* promoter methylation was associated with a clinically relevant HRD phenotype when accompanied by a second hit, typically by LOH. Interestingly, *BRCA1* promoter hypermethylation was observed in melanoma and head and neck cancer, in addition to breast and ovarian cancers, where it has been typically described.[55,56] This suggests that HRD gene promoter methylation may be involved in a broader spectrum of tumors, a clinically important finding, as deficiency in HR is associated with sensitivity to platinum chemotherapies and PARP inhibitor therapies.[77,78]

### Limitations of the study

Coverage for WGTA of tumors using short-read sequencing typically targets 80×–120×, whereas in this study we typically sequenced one PromethION flow cell per sample, for an average of 26×. However, despite this lower coverage, we were able to recover all clinically relevant fusions, phase the majority of somatic small mutations affecting TSGs, and derive a number of complex results from phasing of methylation and expression data.

Phasing of reads in tumors can face challenges from the presence of structural and copy number variation (especially LOH) and may benefit in future from more sophisticated phasing methods such as LongPhase.[79] However, we have shown that this effect is relatively small, most likely due to the presence of both haplotypes in reads derived from the non-tumor cells present in a typical biopsy.

At this time, software for calling somatic small variants from nanopore data is highly experimental and unpublished and claims lower accuracy than short-read sequencing. As we already had access to high-quality somatic small variant calls from high-coverage short-read sequence data, we did not explore somatic small variant calling from the long-read data. We anticipate that somatic small variant calling from nanopore sequencing will improve as new algorithms are developed, as the platform continues to improve in accuracy, and with higher sequence coverage.

This study used the ONT R9 pore sequencing chemistry. The ONT platform continues to rapidly iterate, with improvements in pore structure, library preparation, and base calling. We antic-

ipate that the newer R10 pore chemistry will also improve the accuracy of resolution of mechanisms of tumorigenesis, and continued studies with newer versions of the chemistry would be of benefit to the community.

### Conclusion

In conclusion, we present a comprehensive cohort of tumors sequenced on the nanopore platform. Our initial findings suggest a role for long-read sequencing in personalized cancer medicine through the phasing of somatic mutations, deconvolution of structural variation, identification of tumors driven by HRD, and discovery of allele-specific methylation of cancer genes. The single long-read approach has advantages for clinical use, including providing both sequence and methylation information with comparatively simple library preparation and rapid turnaround times.[80] Additional tumor long-read sequencing is warranted as a complement to the established short-read sequencing paradigm to understand its use in biomarker-driven clinical trials and identifying targeted treatment options. We provide this dataset, complemented by clinical information and short-read sequencing data, as a valuable resource for benchmarking, annotation, and fostering continuous improvement in cancer research and clinical practice.

### RESOURCE AVAILABILITY

#### Lead contact

Requests for further information should be directed to and will be fulfilled by the lead contact, Kieran O'Neill (koneill@bcgsc.ca).

#### Materials availability

There are restrictions on the availability of patient samples due to consent constraints on sending samples outside the REB-approved institution.

#### Data and code availability

- Genomic and transcriptomic sequence datasets for long-read and short-read platforms have been deposited at the European Genome-Phenome Archive (EGA; https://ega-archive.org/) as part of study EGA: EGAS00001001159, as listed in the key resources table, with accession numbers as listed in Table S1. Processed methylation frequency data are available from GEO under accession no. GEO: GSE270257. Other processed data from Long-Read POG, figure source data, and accompanying short-read variants can be downloaded from https://www.bcgsc.ca/downloads/nanopore_pog/. Data on mutations, copy changes, and expression from tumor samples in the POG program are also accessible from https://www.personalizedoncogenomics.org/cbioportal/. WGBS data, ENCODE accessions, and samples from GEO: GSE186458 that were used as normal tissues are listed in the key resources table and included in Table S8.
- Code used to generate figures in this article is available in containerized, reproducible form at https://github.com/bcgsc/long_read_pog (https://doi.org/10.5281/zenodo.13180584).
- Any additional information required to reanalyze the data reported in this paper is available from the lead contact upon request.

### ACKNOWLEDGMENTS

This work would not be possible without the participation of our patients and families, the POG team, Canada's Michael Smith Genome Sciences Centre technical platforms, the generous support of the BC Cancer Foundation and their donors, the Terry Fox Research Institute Marathon of Hope Cancer Centres Network, and Genome British Columbia (project B20POG). We acknowledge contributions from Genome Canada and Genome BC (projects

## Resource

202SEQ to M.A.M. and S.J.M.J., 212SEQ to M.A.M. and S.J.M.J., and 12002 GBC to M.A.M., S.J.M.J., and J.L), Canada Foundation for Innovation (projects 20070 to M.A.M. and S.J.M.J., 30981 to M.A.M., S.J.M.J., and J.L., 30198 to M.A.M., 33408 to M.A.M. and S.J.M.J., 40104 to M.A.M. and S.J.M.J., 42362 to S.J.M.J.), including the CGEN platform (35444 to S.J.M.J.), and the BC Knowledge Development Fund. We acknowledge the generous support of the CIHR Foundation grants program (FDN 143288 to M.A.M.) and the Canadian Institutes of Health Research and Canadian Cancer Society Research Institute (174705, 707104). S.M. and V.P. were the recipients of CIHR Frederick Banting and Charles Best Canada Graduate Scholarships FBD-187583 and GSD-152374, respectively. This study was conducted with the financial support of The Terry Fox Research Institute and the Terry Fox Foundation. The views expressed in the publication are the views of the authors and do not necessarily reflect those of the Terry Fox Research Institute or the Terry Fox Foundation. The results published here are in part based upon analyses of data generated by the following project and obtained from dbGaP (http://www.ncbi.nlm.nih.gov/gap): Genotype-Tissue Expression (GTEx) Project, supported by the Common Fund of the Office of the Director of the National Institutes of Health (https://commonfund.nih.gov/GTEx).

## AUTHOR CONTRIBUTIONS

Wrote the manuscript, K.O.N., E.P., J.F., V.A., G.C., K.D., V.C., S.M., V.P., and A.G.; carried out analyses and prepared figures, K.O.N., E.P., J.F., V.A., G.C., K.D., V.C., S.M., V.P., A.G., and J.H.D.; carried out broader/supporting analyses, R. Corbett, J.H., R.B., D.C., T.W., C.F., Y.S., L.F.P., and E.L.; developed, coordinated, and carried out custom lab work, P.P. and D.E.S.; supervised the work, K.O.N., E.P., F.J.S., J.M.T.N., E.C., K.L.M., R.A.M., R. Coope, A.J.M., M.K.M., L.M.W., K.A.S., S.Y., M.A.M., J.L., and S.J.M.J.

## DECLARATION OF INTERESTS

The following authors disclose relevant potential competing interests: K.O.N., V.P., L.F.P., K.D., J.L., and S.J.M.J. received travel funding from Oxford Nanopore Technologies to present at conferences in 2022 and 2023.

## STAR★METHODS

Detailed methods are provided in the online version of this paper and include the following:

- KEY RESOURCES TABLE
- EXPERIMENTAL MODEL AND STUDY PARTICIPANT DETAILS
- METHOD DETAILS
  - Sample selection and clinical data analysis
  - Sample preparation and sequencing
  - Short-read data analysis
  - Structural variation characterization
  - Viral integration
  - Phasing
  - Allele specific expression
  - Methylation analysis
  - HRDetect and HR gene promoter methylation
  - Extrachromosomal DNA characterization
- QUANTIFICATION AND STATISTICAL ANALYSIS
- ADDITIONAL RESOURCES

## SUPPLEMENTAL INFORMATION

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

## STAR★METHODS

### KEY RESOURCES TABLE

| REAGENT or RESOURCE | SOURCE | IDENTIFIER |
|---|---|---|
| **Biological samples** | | |
| In-house sequenced human tumor and normal samples; see Table S1 | This study; Pleasance et al.[2] | N/A |
| **Critical commercial assays** | | |
| NEBUltra-II kit | New England Biolabs | Cat#E7646A |
| **Deposited data** | | |
| Long-read sequence bams | This study | EGA: EGAS00001001159 |
| Raw methylation data | This study | GEO: GSE270257 |
| Methylation normal tissue data, see Table S8 | Bernstein et al.,[81] ENCODE,[82] and Loyfer et al.[83] | GEO: GSE186458 |
| **Software and algorithms** | | |
| Minimap2 (2.15) | Li[84] | https://github.com/lh3/minimap2 |
| Ploidetect (1.3, 1.4.2) | https://www.biorxiv.org/content/10.1101/2021.08.06.455329v1 | https://github.com/lculibrk/Ploidetect |
| Strelka2 (2.9.10) | Kim et al.[85] | https://github.com/Illumina/strelka |
| Mutect2 (GATK 4.2.0.0) | https://www.biorxiv.org/content/10.1101/861054v1 | https://gatk.broadinstitute.org/hc/en-us/articles/360037593851-Mutect2 |
| ABySS (1.3.4) | Simpson et al.[86] | https://github.com/bcgsc/abyss |
| *Trans*-ABySS (1.4.10) | Simpson et al.,[86] Birol et al.[87] | https://github.com/bcgsc/transabyss |
| Manta (1.6.0) | Chen et al.[88] | https://github.com/Illumina/manta |
| Delly (0.8.7) | Rausch et al.[89] | https://github.com/dellytools/delly |
| MAVIS (2.2.1; 3.1.0) | Reisle et al.[90] | https://github.com/bcgsc/mavis |
| SNPEff (5.0) | Cingolani et al.[91] | https://pcingola.github.io/SnpEff/ |
| BioBloomTools (2.0.11b) | Chu et al.[21] | https://github.com/bcgsc/biobloom |
| STAR (v2.5.2b-XS and Sambamba 0.7.1; 2.7) | Dobin et al.[92] | https://github.com/alexdobin/STAR |
| RSEM (1.3.0) | Li & Dewey[93] | https://github.com/deweylab/RSEM |
| CIBERSORT (1.6.2) | Newman et al.[94] | https://cibersortx.stanford.edu/ |
| SAVANA (1.0.3) | | https://github.com/cortes-ciriano-lab/savana |
| nanomonSV (0.5.0) | Shiraishi et al.[20] | https://github.com/friend1ws/nanomonsv |
| CuteSV (1.0.12) | Jiang et al.[95] | https://github.com/tjiangHIT/cuteSV |
| Sniffles (2.0.7; 1.0.12) | Sedlazeck et al.,[96] Smolka et al.[97] | https://github.com/fritzsedlazeck/Sniffles |
| RepeatMasker (4.1) | Smit et al.[98] | https://github.com/rmhubley/RepeatMasker |
| Custom Viral Integration Workflow | https://doi.org/10.1101/2023.11.04.564800 | https://github.com/vanessa-porter/callONTIntegration |
| BEDTools (2.30.0, 2.23) | Quinlan & Hall[99] | https://github.com/arq5x/bedtools2 |
| IMPALA | https://www.biorxiv.org/content/10.1101/2023.09.11.555771v1 | https://github.com/bcgsc/IMPALA |
| SnpSift (5.1day) | Cingolani et al.[91] | https://github.com/pcingola/SnpSift |
| MBASED (1.34) | Mayba et al.[100] | https://bioconductor.org/packages/release/bioc/html/MBASED.html |
| Bcftools (1.15) | Danecek et al.[101] | https://samtools.github.io/bcftools/bcftools.html |
| MEME suite (5.4.1) | Grant et al.[102] | https://meme-suite.org/meme/ |

*(Continued on next page)*

*Continued*

| REAGENT or RESOURCE | SOURCE | IDENTIFIER |
|---|---|---|
| NanoMethPhase (1.2.0) | Akbari et al.[103] | https://github.com/vahidAK/NanoMethPhase |
| DSS (2.46) | Akbari et al.,[103] Park & Wu[104] | https://www.bioconductor.org/packages/release/bioc/html/DSS.html |
| SignIT (RC1) | | https://github.com/eyzhao/SignIT |
| AmpliconArchitect (1.2) | Deshpande[58] | https://github.com/virajbdeshpande/AmpliconArchitect |
| CNVkit (0.9.10.dev0) | Talevich[105] | https://github.com/etal/cnvkit |
| CycleViz (0.1.2) | | https://github.com/AmpliconSuite/CycleViz |
| Circos (0.69.9) | Krzywinski et al.[106] | https://circos.ca/ |
| IGV (2.14.1) | Robinson et al.[107] | https://igv.org/ |
| Annotatr (1.16.0) | Cavalcante & Sartor[108] | https://bioconductor.org/packages/release/bioc/html/annotatr.html |
| Custom plotting code in R and Python | This study | https://github.com/bcgsc/long_read_pog |
| Other | | |
| Raw and processed data | This study | https://www.bcgsc.ca/downloads/nanopore_pog/ |

## EXPERIMENTAL MODEL AND STUDY PARTICIPANT DETAILS

This study uses patient samples from the POG program, registered under clinical trial number NCT02155621, approved by the University of British Columbia – BC Cancer Research Ethics Board (H12–00137, H14–00681, H20-02317), and approved by the institutional review board. POG IDs ("POGXXXX") were assigned to each case as deidentified codes, the identity of which were known only to the research group. Consent age range and patient sex is included in Table S1. Information on race, ethnicity and ancestry is not explicitly collected as part of the POG program and is therefore not available for this study.

## METHOD DETAILS

### Sample selection and clinical data analysis

As described in Pleasance et al.,[6] patients were referred by their treating oncologist in British Columbia, Canada, and enrolled based on criteria including locally advanced or metastatic cancer, predominantly having received one or more lines of therapy in the metastatic setting, ECOG $\leq 1$, life expectancy >6 months, and the ability to undergo biopsy procedures. Samples with availability of sufficient nucleic acid material after short-read sequencing, were considered for nanopore sequencing. Additionally, samples were preferred if estimated tumor content (excluding surrounding cells such as normal and immune) was >40%, which was true for 97% of samples (Table S1). Tumor tissue was collected as snap-frozen material and the matched normal was blood in all cases except for one hematologic malignancy (POG1022), for which the normal was a matched skin sample.

Overall survival was evaluated from the date of advanced disease diagnosis, defined as the date of incurable, advanced or metastatic disease as determined by radiology or clinical finding if progression was documented with subsequent imaging, whichever was earlier. Kaplan-Meier survival analysis was performed as of August 1, 2023 using the R packages survival (v2.42.3) and survminer (v0.4.2).

### Sample preparation and sequencing
#### Extraction and size selection

Nucleic acids for this study were obtained from previously purified samples from tissues accompanied by tumor estimates as described in Pleasance et al.[2] Purification was performed with an A-Line Evo-pure kit automated on a Hamilton Nimbus96 robot. The overall workflow and automated steps are shown in Figures 1C and S1B. Briefly, frozen tissue sections were immersed in 420 μL of RLT Plus buffer (QIAGEN) containing tris(2-carboxyethyl)phosphine (a reducing agent; TCEP) and a unique sample tracking DNA plasmid, and gently agitated overnight at room temperature. Lysates were transferred from 2 mL tubes to wells of a 1.2 mL plate (Thermo Scientific, AB1127) to which was added 400 mL of 5x bind buffer (80 mL beads in 320 mL isopropyl alcohol). Following a 5 min incubation at room temperature lysates were cleared on a Magnum FLX magnet place (Alpaqua Inc) for 6 min and the protein-containing supernatant removed. The beads, with bound nucleic acids, were washed by pipetting 10 times in wash buffer and returned to the magnet. Beads were washed three times in 70% ethanol then dried for 10 min. 40 mL nuclease-free water

was added to the dried beads and returned to the magnet. The eluted total nucleic acids were transferred to a 96-well storage plate and aliquots taken for quantification using Invitrogen Qubit 4 Fluorometer (Thermo Fisher Q33226).

For samples with concentrations below 166 ng/μL or in volumes greater than 30 μL, 5000 ng of Total Nucleic Acid (TNA) was transferred to a 1.5 mL DNase/RNase free tube and concentrated without heat on the Savant SpeedVac Plus (SC210A) to a maximum volume of 30 μL. This 5000 ng sample in 30 μL was topped up with 10 μL of Qiagen Buffer EB (Cat.19086). The samples were then run on the SAGE Science Blue Pippin instrument using the High Pass Plus Cassette (BPPLUS 10) with a maximum of 4 samples per run. The start range was selected at 15,000 base pairs and the end range was selected at 150,000 base pairs yielding a targeted size of 82,500 base pairs. Following the run completion, each sample was eluted in 80 μL and each sample elution well was then washed with 80 μL of 10 mM Tris, 1mM EDTA containing 0.1% Tween to maximize sample recovery. The total volume of 160μL was then concentrated without heat to 50 μL using the SpeedVac Plus. Each sample was then quantified using the Invitrogen Qubit 4 Fluorometer (Q33226) and normalized to 2000 ng in 47.5 μL for PromethION genome sequencing.

### Library construction and sequencing
Library construction and sequencing followed the Oxford Nanopore Technologies Genomic DNA by Ligation (SQK-LSK110) protocol. DNA libraries starting with 2 μg of sample per library. No shearing was performed. The NEBUltra-II kit, (New England Biolabs, Ipswich, MA, USA, cat. no. E7646A) was used for end-repair and A-tailing. NEBNext quick ligase (E6056S) was used to ligate the Oxford Nanopore sequencing adapter. A final size selection of 0.4:1 ratio beads:library (PCRClean – DX Aline Biosciences L/*N* 06180316) was done to select against smaller molecules. These library preparation steps were performed on Hamilton Nimbus96 liquid handlers. An example deck layout, in this case for the bead purification step, is shown in Figure S1C. DNA libraries were loaded in R9.4.1 pore flow cells on PromethION 24 instrument running software version 19.06.9 (MinKNOW GUI v4.0.23). Sequencing was carried out for 72 h. DNase I (Invitrogen cat no. AM2222) nuclease flush was performed after 24–48 h by reloading the flow cell with the same library mix.

### Basecalling and primary analysis
Basecalling was performed using the guppy basecaller from Oxford Nanopore Technologies, using the "super-accurate" model. Primary analysis was carried out using a NextFlow workflow, which is provided at https://github.com/bcgsc/long_read_pog. Small variants were called using clair3[109] (v0.1-r8) and phased using WhatsHap as included with clair3. Structural variants were called using sniffles[96,97] (1.0.12b) and cuteSV[95] (1.0.12). Methylation (5-mC) was called using nanopolish[13] (0.13.3) and phased using nanomethphase.[103]

### Short-read data analysis
Short-read data from the Illumina platform was generated as described in Pleasance et al.[2] Reads were aligned to the human reference genome (hg38) using Minimap2[84] (v2.15). Regions of copy number variation and losses of heterozygosity were identified using Ploidetect (https://www.biorxiv.org/content/10.1101/2021.08.06.455329v1) (v1.3.0 and v1.4.2). Tumor content (estimated proportion of DNA derived from tumor cells vs. normal cells in the sample) and average ploidy observed in the sequenced tumor were determined based on manual review of Ploidetect results, copy number plots and allelic ratios. Two measures of copy number complexity were computed: the fraction of the genome falling in non-ploidy copy segments, and the genome complexity which is the arithmetic mean of the fraction non-ploidy and the fraction of the total genome size falling in non-ploidy segments, computed based on Ploidetect copy number results with segments less than 10kb merged. Somatic single nucleotide variants (SNVs) and small insertions and deletions (indels) were identified using Strelka2[85] (v2.9.10) and Mutect2 (https://www.biorxiv.org/content/10.1101/861054v1) (in GATK v4.2.0.0). Events assigned PASS by both callers were included, as well as indels called by Strelka2 only with QSS>=50. Tumor mutation burden (TMB) was computed as total called somatic SNVs and indels per megabase. Somatic structural variants (SVs) in DNA data were identified using assembly-based tools ABySS (v1.3.4)[86] and Trans ABySS (v1.4.10)[86,87] and alignment-based tools Manta (v1.6.0)[88] and Delly (v0.8.7),[89] with consensus calls merged using MAVIS[90] (v2.2.1). SVs were filtered to exclude those with identical genomic breakpoints in multiple samples, to remove from the somatic call set germline variants and some technical artifacts. SV events were considered high quality (HQ) if they were called by more than one tool and if a contig could be assembled that aligned across the candidate genomic breakpoint, otherwise they were classed as low quality (LQ). Variants were annotated to genes using SNPEff[91] (v5.0) with the Ensembl database[110] (v100). MSI samples were identified with MSIsensor[111] (v2.0.1). Microbial detection was performed using BioBloomTools (v2.0.11b).[21]

RNA-Seq reads were aligned using STAR[92] (v2.5.2b-XS and Sambamba 0.7.1) and expression was quantified using RSEM[93] (v1.3.0) based on gene models from Ensembl v100. Immune cell deconvolution of RNA-Seq TPM data was performed with CIBERSORT[94] using absolute scores without quantile normalization.

### Structural variation characterization
We conducted two distinct analyses. We performed tumour-only SV calling for all tumors (n = 189). For the subset of tumors with matched normal, (n = 43), we performed somatic SV calling. A literature review was conducted as of May 2023 of existing long reads somatic SV callers, and callers were selected based on the criteria that they had detailed documentation and were continually being maintained over the last year. We identified two somatic SV callers meeting these criteria: SAVANA(v1.0.3) (https://github.com/cortes-ciriano-lab/savana) and nanomonSV(v0.5.0).[20] CuteSV(v1.0.12)[95] and Sniffles(v2.0.7)[96,97] were used as germline SV callers for the tumour-only analysis. Callers were run with default parameters and a minimum size threshold of 50 bp. Intrachromosomal

breakend notation for SAVANA calls were transformed to different SV type calls according to VCF4.2 conventions. Passing SAVANA and nanomonSV events fulfilling a minimum tumor variant allele frequency (VAF) of 0.05 with no event support in the matched normal were subsequently used for downstream analysis.

Post-processing of SVs was conducted with MAVIS (v.3.1.0).[90] We filtered SVs found in the sex and unknown chromosomes, collapsed duplicate SVs, and merged/clustered SVs by breakpoint proximity (100 bp) and type. Insertion end coordinates were calculated by adding the length of the inserted sequence alongside the confidence interval range. SVs called somatic but appearing in paired normals were categorised as false positives and filtered out. SVs were annotated using events from the Database of Genomic Variants (DGV)[112] and frequent events seen in normal WGS using MAVIS. SVs were also annotated based on RepeatMasker (v4.1).[98]

Events were considered to match known fusions if the SV had both breakpoints within 10 bases upstream or downstream of the reported breakpoints and were classified as the same SV type. Events selected for manual review were those that overlap a gene found in OncoKB[113] and which did not have breakpoints overlapping within a repetitive element in RepeatMasker. To visualise somatic events in regions of interest, we took the SAVANA SV calls alongside the Ploidetect(v1.3.0) copy number calls into ShatterSeek.[114]

Corresponding short-read somatic MAVIS post-processed SVs from BreakDancer, DeFuse, Manta, Delly, *Trans*-ABySS, and ChimeraScan were compared to nanopore SVs. Somatic cohort wide level calls were analyzed for any SVs spanning coding elements of oncogenic genes within OncoKB. SVs were considered similar if they clustered within 100 bp of each other. To resolve complex events, we pulled out all reads surrounding the region of the hypothesised event, and took the majority of the reads which supported a certain interpretation. Afterward, we conducted a local assembly of the reads involved in the structural variants (determined by all reads that support an event in the proximal distance of the complex event. Finally, we assessed whether the regions by subsetting for HQ Illumina events and nanopore events, look for those which predict a non-synonymous coding change from short reads and those events which overlap known tumor suppressors or oncogenes. We manually reviewed the transcriptomic data for evidence of irregular splicing patterns and expression profiles through sashimi plots.

### Viral integration

HPV viral breakpoints were detected as described by Porter et al.[115] and using the workflow from GitHub (https://github.com/vanessa-porter/callONTIntegration). Briefly, Sniffles (1.0.12) was used to call breakpoints as translocations between the human chromosomes and the HPV genomes using the following specifications to maximize the accuracy for detected HPV integration breakpoints: max_distance = 50, max_num_splits = −1, report_BND, num_reads_report = −1, min_support = 5, and min_seq_size = 500. Breakpoints were iteratively grouped together into HPV integration events if they had one or more reads that overlapped between the breakpoints as indicated in the VCF, or mapped within 500 kb of each other as measured using BEDTools (v. 2.30.0).[99] The first condition ensured that breakpoints appearing distant to each other relative to the reference genome but were physically linked through fusion rearrangements, could be paired together. The second condition ensured breakpoints mapping near each other but lacked a read long enough to link them together. The collection of HPV breakpoints that were grouped together through these two methods were referred to as an integration event. All read names belonging to an integration event were retained for later analyses. Integration event structures were determined using the read alignment patterns as described by Porter et al., 2023 (https://doi.org/10.1101/2023.11.04.564800). The collection of HPV breakpoints that were grouped together through these two methods were referred to as an integration event. All read names belonging to an integration event were retained for later analyses. Integration event loci were defined as the integration breakpoints within an event that map within 500 kb of each other, as determined using BEDTools (v. 2.30.0). Therefore, integration events spanning multiple chromosomes or large genomic expanses would have multiple integration event loci. The integration event loci were used for regional analyses such as determining neighboring genes. The multi-breakpoint event was analyzed using the workflow found here: https://github.com/vanessa-porter/comSVis, which sectioned the event using the collection of all SV breakpoints that were phased within the event. The mean depth between the breakpoints was then calculated using BEDTools (v. 2.30.0) for visualisation.

### Phasing

Individual phasing statistics were calculated for each phased VCF using WhatsHap stats. Read N50 is the length at which reads of the same or greater length represent 50% of the genome. To estimate the phasing rate across tumor suppressor genes, we determined the number of protein-coding genes (GENCODE v43) that are contained within a single phase block for each sample using bedtools intersect, restricting overlapping genes to those that had a 100% overlap with a given phase block. Putative biallelic somatic variants with potential biological or clinical significance were identified from the POG genomic reports. Tumor suppressor genes were defined by the COSMIC Cancer Gene Census. Long reads were colored by predicted haplotype using WhatsHap haplotag, and all candidate biallelic variants were manually reviewed in IGV.

### Allele specific expression

The IMPALA pipeline (https://github.com/bcgsc/IMPALA; https://www.biorxiv.org/content/10.1101/2023.09.11.555771v1) was used to detect ASE genes in the POG cohort (n = 172), which uses tumor RNA-seq data and phased variants generated from tumor long reads. STAR aligner[92] (v2.7) is used to align the RNA reads to the genome before performing variant calling using Strelka (v2.9).

This generates allelic read counts for each variant. Heterozygous SNPs are filtered for and annotated with haplotype information and gene annotation using the phased variants and SnpEff[91] (v5.0) respectively. SnpSift (v5.1d) formats the allelic read count and annotations as preprocessing for ASE detection. RSEM[93] (v1.3) is used to quantify expression of RNA-seq data and filter out genes with expression lower than 1 TPM.

MBASED[100] (v1.34) is the main software used to calculate ASE. Biallelic genes are expected to have an allelic read count ratio of 0.5. MBASED performs a beta-binomial test on each phased SNP to assess the statistical deviation away from the expected 0.5 ratio. Afterward, MBASED utilises meta-analysis with haplotype information to aggregate SNP-level data to gene-level major allele frequency data. P-value is adjusted with Benjamini-Hochberg method. Genes with major allele frequency above 0.65 and adjusted *p*-value below 0.05 are classified as allelically expressed.

Post-processing of the allele specific expression is done by integrating additional information to determine the potential cause of the allele specific expression. CNV data from Ploidetect, allelic methylation data from NanoMethPhase and somatic variant calls can be used as optional inputs for IMPALA software. Bedtools (v2.23) intersect is used to intersect ASE data with CNV states, allelic methylation and somatic calls. Additionally, SnpEff is used to annotate and filter for nonsense variants in ASE gene as a potential genetic mechanism. Lastly, bcftools (v1.15)[101] consensus is used to generated consensus sequence of both allele based on the phased variants and FIMO from the MEME suite (v5.4.1)[102] detects transcription factor binding sites on both alleles and find differences between the allele. Disruption of transcription factor binding sites could lead to ASE. The final output of the workflow is a summary table with allelic information in addition to the *cis*-acting elements which can be used for downstream analysis to identify genes of interest.

### Methylation analysis

For non-allelic methylation analysis we used nanopolish methylation frequency results. As normal methylation data, 267 WGBS datasets from various tissue/cell types were gathered from Epigenomics Roadmap,[81] ENCODE,[82] and Loyfer et al.[83] (GSE186458). To analyze overall methylation at different genomic regions, we used bedtools intersect to overlap CpG methylation frequency data to Repeats, TF binding sites (also includes CTCF binding sites), and CGIs from the UCSC table browser,[116] promoters (1500 bp upstream and 500 bp downstream of TSS in Ensembl100 transcripts GRCh38.p13[110]), 500bp up and downstream of polyA sites from PolyA_DB 3,[117] and enhancers from GeneHancer v5.14.[118] For tSNE, we used CGIs, Promoters, CTCF binding sites, and Enhancers regions with standard deviation $\geq$ 0.2 of the mean methylation between tumor types or biopsy sites. For both phased and unphased data, if the methylation was at strand level, for each CpG site we aggregated the number of reads as methylated and number of all reads from both strands to calculate consensus methylation frequency.

To detect aDMRs in each sample, the phased haplotype 1 and 2 results from NanoMethPhase were used and DMRs were called in each sample using NanoMethPhase dma module with default options with DSS R package version 2.46.0[103,104]. Detected DMRs were further filtered to keep DMRs with |diff.methyl (delta methylation)| $\geq$ 0.15. To filter detected aDMRs in tumor samples and keep only tumour-specific aDMRs, in addition to ignoring aDMRs that overlap to more than one matched normal sample, we excluded aDMRs showing partial methylation in more than 1% of the normal WGBS samples. Partial methylation is an indication of allelic methylation because only one allele is methylated and overall methylation at the region will be ~50%. To explore partial methylation, for each WGBS sample, we used CpGs with at least five mapped reads and at each detected aDMR we counted the number of CpGs with partial methylation (methylation frequency between 0.35 and 0.65). We also excluded aDMRs overlapping within a 10 kb window of known imprinted genes and regions to exclude aDMRs stemming from imprinting.[119] An aDMR with 0.35–0.65 methylation is then considered partially methylated if it had at least five CpGs in the WGBS sample and more than 60% of the CpGs showed partial methylation. To overlap detected aDMRs to genomic regions we used bedtools intersect -e -f 0.5 -F 0.5. TF enrichment for recurrent genes with aDMR at their promoter was evaluated using the Enrichr ChEA 2022 database[120] (https://maayanlab.cloud/Enrichr/). CIBERSORT version 1.6.2 was used to infer immune infiltrate proportions using gene expression data.

### HRDetect and HR gene promoter methylation

We used HRDetect, a tool which aggregates different mutational signatures including single base substitution signatures, structural variant signatures and microhomology-mediated deletions, to predict HRD in our samples. HRDetect scores were computed from short-read sequencing data using a logistic regression model with the same intercept and coefficients as those reported in the previously trained model, without adjustment.[121] The intercept was −3.364 and the coefficients were 1.611, 0.091, 1.153, 0.847, 0.667, and 2.398, respectively, for the six HRD signatures: (i) SBS3, (ii) SBS8, (iii) SV signature 3, (iv) SV signature 5, (v) the HRD index, and (vi) the fraction of deletions with microhomology. The contribution of previously reported mutational signatures in the Catalog of Somatic Mutations in Cancer (COSMIC v3.1, https://cancer.sanger.ac.uk/cosmic/signatures) was calculated using Monte Carlo Markov Chain (MCMC) sampling (https://github.com/eyzhao/SignIT). Short-read MAVIS calls that were detected by more than one tool and for which the contig could be assembled were included in the analysis and the contribution of the previously reported SV mutational signatures was calculated using MCMC sampling (https://github.com/eyzhao/SignIT).[122] The HRD index was computed as the arithmetic sum of loss of heterozygosity, telomeric allelic imbalance, and large-scale state transitions scores. The microhomology fraction was determined as the proportion of deletions which were larger than three base pairs and demonstrated overlapping microhomology at the breakpoints.[77] All signatures were log transformed and normalized so that each feature had a mean of 0 and standard deviation of 1.[121]

Promoter methylation of the following HR genes, selected based on their established roles in homologous recombination repair, were investigated to examine associations with high HRDetect score: *BARD1, BLM, BRCA1, BRCA2, BRIP1, DNA2, EXO1, MRE11A, NBN, PALB2, RAD50, RAD51, RAD51B, RAD51C, RAD51D, RAD52, RAD54L, RBBP8, WRN, XRCC2, XRCC3, ATM, BAP1, CUL3, EME1, ERCC1, ERCC4, FBXO18, GEN1, HELQ, MUS81, PARPBP, PCNA, POLD1, POLK, POLN, PSIP1, RAD51AP1, RECQL5, RIF1, RMI1, RMI2, RPA1, RPA2, RPA3, RTEL1, SLX1A, SLX4, TOP3A, TP53BP1,* and *USP11*. Promoter is defined as 1500 bp upstream and 500 bp downstream of TSS. Methylation frequencies from Nanopolish in the promoter regions of 51 HR genes in the tumors and in publicly available matched tissues (https://epigenomesportal.ca/ihec/) were compared. 'Methylated' site in the promoter is defined as that methylation of the site being >1 SD from the mean normal methylation level in the matched tissue and then the fraction of methylated sites in the gene promoter is computed.

### Extrachromosomal DNA characterization

To identify potential ecDNAs from short-read WGS data, we ran PrepareAA (v0.1203.1) with CNVkit[105] (v0.9.10.dev0) for copy number calling followed by AmpliconArchitect[58](v1.2) with default settings (–gain 4.5 and –cnsize_min 50000). For ecDNA structure visualization, we first used CycleViz (v0.1.2; github.com/AmpliconSuite/CycleViz) to obtain breakpoints predicted by AmpliconArchitect for ecDNAs with only one predicted substructure, followed by circos[106] (v0.69.9), in which we overlaid additional methylation data obtained from long-read WGS data.

To validate the structure of select ecDNAs predicted by AmpliconArchitect,[58] which uses short-read WGS data, we manually reviewed supplementary reads from the long-read WGS data in IGV[107] (v2.14.1). Specifically, we looked for reads mapping to both sides of each predicted breakpoint +/− 100 bp. We assigned an ecDNA to a specific haplotype based on whether reads mapping to SVs associated with the ecDNA mapped to reads within the haplotype-phased bam file. Specifically, we extracted SVs associated with the ecDNA from the output of AmpliconArchitect, found these SVs in the long-read WGS data from Sniffles (v1.0.12b), and then mapped these reads to both tumor haplotype bam files. For further validation of the assignment of an ecDNA to a specific haplotype, we viewed the ecDNA in IGV to confirm amplified regions co-localized with the ecDNA regions of AmpliconArchitect.

We used annotatr[108] (v1.16.0) in R (v4.0.2) to extract gene promoters overlapping both ecDNA regions and DMRs obtained from the allele-specific methylation pipeline prior to filtering out CNVs (see also Allele-specific methylation). We selected *NRG1* in a breast cancer sample, for further analysis as it is a known cancer gene in breast cancer,[59–62] had multiple DMRs falling within it, including two in promoter regions, and had >0.5 methylation frequency for one haplotype and <0.5 methylation frequency for the other haplotype for each promoter DMR. Plots for *NRG1* methylation were constructed in R with tidyverse[123] (v2.0.0) and patchwork (v1.1.2.9000)[124] functions with ggbio[125] (v1.38.0) and EnsDB.Hsapiens.v86 (v2.99.0) for gene annotation.

We compared *NRG1* and other genes within the *NRG1* pathway (*ERBB2, ERBB3,* and *AKT1*) between the sample containing the *NRG1* ecDNA (n = 1) to other breast cancers within the cohort (n = 39) in terms of RNA expression in TPM. Permutation tests were used to assess significance between the ecDNA sample and the rest of the breast cancers in the cohort using the coin[126] (v1.4-2) package and Bonferroni multiple testing correction. We also reviewed ASE results for *NRG1* in the sample of interest (see also Allele-specific expression).

### QUANTIFICATION AND STATISTICAL ANALYSIS

Statistical analysis was performed using R or Python or using the statistics provided by specific software, with packages and software as noted in the method details section. Statistical details of experiments including tests used, sample size and *p*-values can be found in the results section, figures and/or figure legends. Tests were two-tailed, considered significant at *p* < 0.05 and multiple test corrected as noted.

### ADDITIONAL RESOURCES

Clinical trial registry: https://clinicaltrials.gov/study/NCT02155621.

