## [Document S2. Transparent peer review records for O’Neill et al · Cell Genomics]

Long-read sequencing of an advanced cancer cohort resolves rearrangements, unravels haplotypes, and reveals methylation landscapes

Kieran O'Neill, Erin Pleasance, Jeremy Fan, Vahid Akbari, Glenn Chang, Katherine Dixon, Veronika Csizmok, Signe MacLennan, Vanessa Porter, Andrew Galbraith, Cameron J. Grisdale, Luka Culibrk, John H. Dupuis, Richard Corbett, James Hopkins, Reanne Bowlby, Pawan Pandoh, Duane E. Smailus, Dean Cheng, Tina Wong, Connor Frey, Yaoqing Shen, Eleanor Lewis, Luis F. Paulin, Fritz J. Sedlazeck, Jessica M.T. Nelson, Eric Chuah, Karen L. Mungall, Richard A. Moore, Robin Coope, Andrew J. Mungall, Melissa K. McConechy, Laura M. Williamson, Kasmintan A. Schrader, Stephen Yip, Marco A. Marra, Janessa Laskin, Steven J.M. Jones

Summary

Initial submission: Received : Feb 23, 2024

Scientific editor: Sara Rohban

First round of review: Number of reviewers: 3
Revision invited : Apr 03, 2024
Revision received : Jun 26, 2024

Second round of review: Number of reviewers: 2
Accepted : Sep 18, 2024

Data freely available: YES

Code freely available: YES

This transparent peer review record is not systematically proofread, type-set, or edited. Special characters, formatting, and equations may fail to render properly. Standard procedural text within the editor's letters has been deleted for the sake of brevity, but all official correspondence specific to the manuscript has been preserved.

Referees' reports, first round of review

Reviewer #1:

The authors present the sequencing of 189 tumor samples with Oxford Nanopore Technologies (ONT) sequencing as well as additional short-read genome and RNA-seq. The samples derive from a broad array of cancer types. The authors make the sequencing data available for downstream use from the European Genomics Archive. The dataset looks pretty useful but has a few limitations. Firstly, the ONT data is R9 chemistry, which lowers its accuracy relative to the latest R10 (and now Q27) chemistries. Secondly, only 43 of the samples have matched normal sequencing with long-reads, a significant limitation given the challenges of distinguishing between tumor and germline variants, and particularly SVs in polymorphic loci. Finally, and this is not a criticism, but the data is managed access, and therefore will find limited downstream usage given the barriers to gaining and using the data (I strongly believe the field needs more open data to spur methods development and testing).

Given the R9 nature of the data the authors sensibly focus on the usage of the data for structural variant discovery, phasing and methylation analysis, rather than focusing on small variant detection. I was genuinely really impressed by the analyses presented which are a pretty comprehensive survey of things you might want to do with this long-read data in the cancer space. It may be that the paper has previously been reviewed elsewhere (guessing), but I found the level of polish to be well above most initial submissions. I have a few relatively minor comments that I'd advise the authors address:

- * The figures are mostly beautiful, but I couldn't follow Figure 1 panel C, the legend isn't really useful.
- * The number of SV insertions and deletions in Fig 1A looks rather large, implying 40k ish/genome. I'd suggest some filtering because I'd guess there is some cruft in there. I'd expect closer to 25-30k ish. Maybe the multiple methods used were not well merged?
- * Figure 1f: also pretty inscrutable, tbh.
- * The use of diploid phasing with whatshap is a bit questionable in some regions of the genome where there is LOH or CNVs. What steps were taken to avoid making weird calls in non-diploid regions? (I see nothing in the methods)
- * In the methylation analysis, was there any attempt to deconvolve the normal contamination? If it is described I missed it. I see comments in the discussion, but it might have been useful to exclude samples where substantial immune or normal tissue contamination was present?
- * The study of ec-DNA seems a bit limited, with no attempt to use the long-reads to systematically validate the 76 predicted ecDNAs. This is a shame.

Reviewer #2:

In this manuscript, Kieran O'Neill et al. sequenced a POG cohort comprising 189 patient tumours and 41 matched normal samples using Nanopore long-read sequencing technologies. Taking advantage of the long-read sequencing, they can resolve complex structural variants, perform long-range haplotype-phasing and detect associated allelic expression and methylation. These comprehensive analyses provide some novel insights into the oncogenomics though most analysis finally confined to one or two known cancer-related genes. Anyway, this dataset is a valuable resource for cancer research. I have several comments as follows:

Points;

1. In Figure 2c, I cannot figure out the meaning of I+N (without figure legends). If it is intersection of I and N, panels for Dup and Inv are almost impossible; otherwise, union of I and N is not applicable for panels of Del and Ins. Besides, what does the x-axis represent in the right panel of Figure 2c.
2. For Viral Integration in Methods (p.25), brief method description for structure construction of integration event is desired in addition to indexing another unpublished manuscript.
3. The number described in the main text (19 pairs across 18 cases in p.8) is not matched with Table 1, please check it.
4. The description for PALB2 case (ASE) in the main text (p.10) is also not consistent with that in Figure 3g legends (BAE).
5. It would be better to also show other types of regulatory sites in addition to CGIs (Figure 4c) as Extended Data Figure if possible.
6. It is the best to release the original data as well as processed data.

Minor point;

1. There is a typo "iether" in Figure4 legend.

Reviewer #3:

In this manuscript, the authors performed nanopore sequencing on an advanced cancer cohort, which included 189 tumor samples of various cancer types. Using the data, they conducted comprehensive analyses, such as haplotype phasing, DNA methylation analysis, and detection of structural variations (SVs). For SV detection, the SVs called from long-read data were evaluated by comparing them to those called from short-read data, and they identified some complex SVs. Using long-read data, they confirmed HPV integrations in the samples and verified the effect of the integration on expression of the surrounding genes. Based on the phasing results of somatic mutations, they demonstrated biallelic mutations in some tumor suppressor genes. They also detected allelic-specific expression (ASE) using the short-read RNA-seq dataset obtained in their previous study and compared ASE with allelic differential methylated regions (aDMRs). They observed dysregulation of methylation in oncogenes and DNA repair genes. Finally, they analyzed extrachromosomal DNA and its epigenomic status. The authors have conducted a multifaceted analysis that leveraged the advantages of long reads, uncovering a variety of features that would have been challenging to identify using short-read data alone. Given the scarcity of large datasets for long-read cancers, the dataset generated in this study holds significant value as a resource. Furthermore, this manuscript serves as an important example of cancer analysis using long-read datasets. There are, however, several points to be considered for the possibility of further improvements:

1. The authors have evaluated allelic expression and methylation, but the underlying mechanisms remain unclear. To enhance our understanding, they should focus on mutations within regulatory regions, such as promoters and enhancers, and explore their effects on allelic differential methylated regions (aDMRs) and allelic-specific expression (ASE). Additionally, considering that frameshift mutations and nonsense mutations often lead to premature termination codons, it would be valuable to analyze the impact of premature codons derived from nonsense mutations or frameshift mutations on ASE. Notably, Figure 3f demonstrates lower expression in the allele with the frameshift mutation, suggesting that nonsense-mediated mRNA decay (NMD) might influence allelic expression.
2. In Extended Figure 6, the authors reveal cancer-specific allelic loss of gene body methylation and elevated expression on the other allele in the DUSP22 gene. It would be worthwhile to investigate whether similar relationships between alterations in gene body methylation and gene expression are observed in other genes.
3. In Table 1, the KAT5 gene of POG446 is listed both in trans and in cis. Based on the context, "in cis" appears to be the correct designation.
4. Figure 3A shows that only three samples exhibit longer phased block lengths. It might be worthwhile to explore what this variation is related to (e.g. sequencing depth).
5. Ensuring clear correspondence between figures and Case IDs is essential. For instance, in places like Figure 2f and Supplementary Figure 1, proper annotations in figures or their legends would enhance clarity.

6. On page 12, explaining why "IDH gain-of-function mutations" cause hypermethylation would make it easier for many readers to understand. A brief explanation of the mechanisms behind this effect would improve understanding.
 7. In Extended Figure 7c, the authors demonstrate significant enrichment for PRC1 and PRC2 complexes binding sites in aDMRs. Discussing the biological implications of this finding would provide valuable insights.
 8. In Figure 4g and i, ensuring consistency in abbreviations (IUM and IPU) is essential for clarity
 9. In Figure 5c, 5d, and 5f, HP1(germ) and HP2(germ) appear to represent blood-derived data. It is essential to explicitly state this information in the figure legends.
-

Authors' response to the first round of review

We'd like to thank the reviewers for their enthusiastic and constructive comments! We have detailed our responses in-line below (highlighted in bold). Please note that we have included additional data in response to reviewer comments, and as noted in one of the specific comments below, this has led to our inclusion of one additional author, with permission from co-authors.

Reviewer #1

Reviewer #1: The authors present the sequencing of 189 tumor samples with Oxford Nanopore Technologies (ONT) sequencing as well as additional short -read genome and RNA-seq. The samples derive from a broad array of cancer types. The authors make the sequencing data available for downstream use from the European Genomics Archive. The dataset looks pretty useful but has a few limitations. Firstly, the ONT data is R9 chemistry, which lowers its accuracy relative to the latest R10 (and now Q27) chemistries. Secondly, only 43 of the samples have matched normal sequencing with long-reads, a significant limitation given the challenges of distinguishing between tumor and germline variants, and particularly SVs in polymorphic loci. Finally, and this is not a criticism, but the data is managed access, and therefore will find limited downstream usage given the barriers to gaining and using the data (I strongly believe the fields needs more open data to spur methods development and testing).

Given the R9 nature of the data the authors sensibly focus on the usage of the data for structural variant discovery, phasing and methylation analysis, rather than focusing on small variant detection. I was genuinely really impressed by the analyses presented which are a pretty comprehensive survey of things you might want to do with this long-read data in the cancer space. It may be that the paper has previously been reviewed elsewhere (guessing), but I found the level of polish to be well above most initial submissions. I have a few relatively minor comments that I'd advise the authors address:

Thank you! This was actually our first invitation for revision. We just had a good few rounds of internal revisions before submission.

We fully agree regarding availability of datasets. Unfortunately, whole-genome cancer patient data is difficult to make fully open access for ethical and consent reasons. But we have endeavoured to make as much anonymized data available as possible, including somatic calls, methylation, etc. We also understand that the process of requesting access to the complete controlled dataset via EGA is relatively straightforward, if less convenient than just downloading from the open web.

* The figures are mostly beautiful, but I couldn't follow Figure 1 panel C, the legend isn't really useful.

We're assuming this meant Figure 2 (the SV figure). The Fig 2c legend was one that slipped through during manuscript preparation – we've fixed that, added a label for the x axis, and have otherwise tried to clarify that figure. Please note that this is now Fig 2b due to changes in response to other comments.

* The number of SV insertions and deletions in Fig 1A looks rather large, implying 40k ish/genome. I'd suggest some filtering because I'd guess there is some cruft in there. I'd expect closer to 25-30k ish. Maybe the multiple methods used were not well merged?

We fully agree. It seems that cuteSV was overcalling somewhat, and the numbers shown were taken from the union of cuteSV and sniffles calls. We have replaced the plot with the intersection of the calls between the two callers, which brings the number closer to the 20-25k we might expect. We have included Figure S2 to show this intersection of calls between the callers. We have also moved Figure 2a to the supplementary section, since the focus of the main figure is on somatic SVs.

* Figure 1f: also pretty inscrutable, tbh.

Thank you for your feedback. The visual table in this figure has been simplified, and POG identifiers added to make it more clear that this refers to 5 individual samples. This is now Figure 2e. Part g of the figure (now Figure 2f) has also been reworked to improve readability.

* The use of diploid phasing with whatshap is a bit questionable in some regions of the genome where there is LOH or CNVs. What steps were taken to avoid making weird calls in non-diploid regions? (I see nothing in the methods)

Thanks – this is a very good point. We did look broadly at the effect of genomic rearrangements on phasing (see Figure S7c), and found it to have a significant but fairly weak effect. Curiously, we often found that there was sufficient normal tissue present in the samples sequenced to represent both haplotypes. This enabled phasing even in the presence of LOH (e.g. in some instances of the HRD

promoter methylation, or the phasing of the tumour suppressor double hits). However, we are aware that there are newer phasing tools (such as longphase) which can account for SVs, which we will be testing in future. We have added the below text to the discussion section to address this:

“Phasing of reads can also face challenges from the presence of structural and copy number variation (especially LOH), and may benefit in future from more sophisticated phasing methods such as LongPhase. However, we have shown that this effect is relatively small, most likely due to the presence of both haplotypes in reads derived from the non-tumour cells present in a typical biopsy.”

* In the methylation analysis, was there any attempt to deconvolve the normal contamination? If is described I missed it. I see comments in the discussion, but it might have been useful to exclude samples where substantial immune or normal tissue contamination was present?

This is an important topic, as tumour biopsies from advanced cancers represent a mix of normal tissue, tumour, and immune cell components. In this cohort, we did generally apply a minimum threshold for tumour content for cases to be re-sequenced on the PromethION, such that 97% of cases have an estimated tumour content of >40%. We estimated tumour content of all samples sequenced as part of the POG project, based on the short-read sequencing, using a combination of copy number depth and allelic ratios in tumour DNA data. We have not explored deconvolution and normal cell subtraction from the methylation data at this time, as this is a complex topic, but we have now added details on deconvolution of the immune component from RNA data based on the CIBERSORT algorithm (Table S1); this is accompanied by the inclusion of an additional author who was involved in providing this data (Eleanor Lewis). We have referenced this in the results: “The tumour content ranged from 21-100% (median 67%), with estimates of immune infiltration provided in Table S1.”

Added clarification in the methods:

“Additionally, samples were preferred if estimated tumour content (excluding surrounding cells such as normal and immune) was >40%, which was true for 97% of samples (Table S1).”

And also added the below text to the discussion:

“Examination of the tumour microenvironment can be accomplished by deconvolving cell types from RNA-seq data, and we have included CIBERSORT results created from short-read RNA-seq for this cohort. Similar methods can be applied to 5-mC derived from long-read tumour sequence data, such as MethylCIBERSORT.”

* The study of ec-DNA seems a bit limited, with no attempt to use the long-reads to systematically validate the 76 predicted ecDNAs. This is a shame.

We fully agree that the remaining ecDNAs need to be systematically investigated. The PhD student responsible for that section of the manuscript is working on that as a separate project in its own right,

and we are very much looking forward to seeing (and publishing) the results! To this end, we have added the below text to the discussion:

“Our initial results unravelling the biology of a putative ecDNA in POG816 using long-read sequencing have been promising, suggesting future research could include more comprehensive exploration of the remaining ecDNAs identified in short-read sequencing in this cohort.”

Reviewer #2

Reviewer #2: In this manuscript, Kieran O'Neill et al. sequenced a POG cohort comprising 189 patient tumours and 41 matched normal samples using Nanopore long-read sequencing technologies. Taking advantage of the long-read sequencing, they can resolve complex structural variants, perform long-range haplotype-phasing and detect associated allelic expression and methylation. These comprehensive analyses provide some novel insights into the oncogenomics though most analysis finally confined to one or two known cancer-related genes. Anyway, this dataset is a valuable resource for cancer research. I have several comments as follows:

Points;

1. In Figure 2c, I cannot figure out the meaning of I+N (without figure legends). If it is intersection of I and N, panels for Dup and Inv are almost impossible; otherwise, union of I and N is not applicable for panels of Del and Ins. Besides, what does the x-axis represent in the right panel of Figure 2c.

Thank you – Fig 2c definitely needed work (as both you and reviewer 1 have noted). We've relabeled the bars to “Short-read only” “Long-read only” and “Both platforms” to make this clearer. We have also added a label for the x axis of the size distribution plots, and titles for the subfigures to clarify further. This is now Figure 2b due to changes in response to other comments.

2. For Viral Integration in Methods (p.25), brief method description for structure construction of integration event is desired in addition to indexing another unpublished manuscript.

Thank you. We have expanded the methods text to include greater detail, as follows:

“HPV viral breakpoints were detected as described by Porter et al⁹⁶ and using the workflow from GitHub (<https://github.com/vanessa-porter/callONTIntegration>). Briefly, Sniffles (1.0.12) was used to call breakpoints as translocations between the human chromosomes and the HPV genomes using the following specifications to maximize the accuracy for detected HPV integration breakpoints: max_distance = 50, max_num_splits = -1, report_BND, num_reads_report = -1, min_support = 5, and min_seq_size = 500. Breakpoints were iteratively grouped together into HPV integration events if they

had one or more reads that overlapped between the breakpoints as indicated in the VCF, or mapped within 500 kb of each other as measured using BEDTools (v.

2.30.0). The first condition ensured that breakpoints appearing distant to each other relative to the reference genome but were physically linked through fusion rearrangements, could be paired together. The second condition ensured breakpoints mapping near each other but lacked a read long enough to link them together. The collection of HPV breakpoints that were grouped together through these two methods were referred to as an integration event. All read names belonging to an integration event were retained for later analyses. Integration event structures were determined using the read alignment patterns as described by Porter et al, 2023 (<https://doi.org/10.1101/2023.11.04.564800>). The collection of HPV breakpoints that were grouped together through these two methods were referred to as an integration event. All read names belonging to an integration event were retained for later analyses. Integration event loci were defined as the integration breakpoints within an event that map within 500 kb of each other, as determined using BEDTools (v. 2.30.0). Therefore, integration events spanning multiple chromosomes or large genomic expanses would have multiple integration event loci. The integration event loci were used for regional analyses such as determining neighbouring genes.

The multi-breakpoint event was analysed using the workflow found here:

<https://github.com/vanessa-porter/comSVis>, which sectioned the event using the collection of all SV breakpoints that were phased within the event. The mean depth between the breakpoints was then calculated using BEDTools (v. 2.30.0) for visualisation. ”

3. The number described in the main text (19 pairs across 18 cases in p.8) is not matched with Table 1, please check it.

Thanks, this is well-spotted. We have fixed the duplicated event in the table, and ensured the numbers in the text are correctly matched. We also added a footnote to the table highlighting the case that appears in the list more than once.

4. The description for PALB2 case (ASE) in the main text (p.10) is also not consistent with that in Figure 3g legends (BAE).

Thank you for the attention to detail, the text has been updated to reflect that PALB2 is BAE not ASE. In the same paragraph, we also moved the text reference to the figure to a clearer position and added a reference to Table S1 where the HRD data is found. We have amended the text to read:

“POG976 with both PALB2 somatic and germline variants, confirmed by phasing to be opposite alleles, shows BAE as both alleles are impacted by truncating events (Figure 3g). Consistent with loss of function of PALB2, this cholangiocarcinoma was characterised by strong mutational signatures of HRD (Table S1).”

5. It would be better to also show other types of regulatory sites in addition to CGIs (Figure 4c) as Extended Data Figure if possible.

Absolutely! We have added Figure S10 showing a range of other regulatory regions.

6. It is the best to release the original data as well as processed data.

The original bam files are in EGA, but we do note that we had not included the location of the original methylation data in the manuscript as submitted. We have now noted the location in GEO accession of the methylation data (GSE270257) under the data availability section. We have also ensured that the DMR calls themselves have been included on the data hosting site.

Minor point;

1. There is a typo "iether" in Figure4 legend.

Response: thank you, this has been fixed.

Reviewer #3

Reviewer #3: In this manuscript, the authors performed nanopore sequencing on an advanced cancer cohort, which included 189 tumor samples of various cancer types. Using the data, they conducted comprehensive analyses, such as haplotype phasing, DNA methylation analysis, and detection of structural variations (SVs). For SV detection, the SVs called from long-read data were evaluated by comparing them to those called from short-read data, and they identified some complex SVs. Using long-read data, they confirmed HPV integrations in the samples and verified the effect of the integration on expression of the surrounding genes. Based on the phasing results of somatic mutations, they demonstrated biallelic mutations in some tumor suppressor genes. They also detected allelic-specific expression (ASE) using the short-read RNA-seq dataset obtained in their previous study and compared ASE with allelic differential methylated regions (aDMRs). They observed dysregulation of methylation in oncogenes and DNA repair genes. Finally, they analyzed extrachromosomal DNA and its epigenomic status.

The authors have conducted a multifaceted analysis that leveraged the advantages of long reads, uncovering a variety of features that would have been challenging to identify using short-read data alone. Given the scarcity of large datasets for long-read cancers, the dataset generated in this study holds significant value as a resource. Furthermore, this manuscript serves as an important example of cancer analysis using long-read datasets. There are, however, several points to be considered for the possibility of further improvements:

1. The authors have evaluated allelic expression and methylation, but the underlying mechanisms remain unclear. To enhance our understanding, they should focus on mutations within regulatory regions, such as promoters and enhancers, and explore their effects on allelic differential methylated regions (aDMRs) and allelic-specific expression (ASE). Additionally, considering that frameshift mutations and nonsense mutations often lead to premature termination codons, it would be valuable to analyze the impact of premature codons derived from nonsense mutations or frameshift mutations on ASE. Notably, Figure 3f demonstrates lower expression in the allele with the frameshift mutation, suggesting that nonsense-mediated mRNA decay (NMD) might influence allelic expression.

To explore regulatory mutations as suggested, we have now included an additional analysis of mutations in regions previously reported as promoter hotspots and examined them for ASE and aDMRs. We have included information about these mutations in the new Table S6. Unfortunately, recurrent promoter mutations are rare and therefore there were insufficient samples in our cohort for most hotspots to make any clear statements. However, TERT promoter mutations were the most frequent, and we have now included an analysis on associations with expression, ASE, and allele-specific methylation. We have added Figure 5e-f, Figure S12c, and added the following text to describe these results:

“We further examined whether known promoter mutations are associated with ASE and methylation. We examined the most frequent hotspot noncoding mutations described in the POG570 cohort², which are associated with the genes TERT, PLEKHS1, ADGRG6, and AP2A1 (Table S6). As these mutations are still rare, in the long-read POG cohort we found only four to five mutations associated with the described hotspot regions in each of PLEKHS1, ADGRG6, and AP2A1, and with such few samples were unable to find associations with ASE or aDMR regions. TERT promoter mutations were found in a total of 13 samples. TERT, an oncogene encoding telomerase associated with maintaining telomeres in rapidly dividing cancer cells, is frequently overexpressed in tumours⁴⁹. Mutations in the long-read POG cohort were indeed associated with higher TERT gene expression ($p < 0.01$, Figure 5e). As a transcription factor, TERT generally has low TPM which makes measurement of ASE more challenging. However, examining the allele-specific data, we found more ASE for the allele containing the mutations ($p < 0.006$) in part due to the higher expression bringing more samples above the $TPM \geq 1$ threshold used for calculating ASE (see Methods). We also found no cases where TERT had statistically significant BAE. The TERT promoter is normally unmethylated⁴⁹, which is also observed in long-read POG blood normal samples (Figure S12c). We found that TERT promoter mutations frequently overlapped tumour aDMRs, but interestingly tended to be found on the less methylated allele ($p = 0.13$, Figure 5f). In tumour cells, the TERT promoter is reported to be frequently methylated, but there has been some debate about whether TERT mutations and promoter methylation are mutually exclusive and this has been explored in cell lines⁵⁰. Our allele-specific data shows that TERT promoter mutation and promoter methylation may indeed co-occur in a single patient tumour, but that these two alterations may be on different alleles.”

Your point about nonsense mutations and NMD leading to ASE is a good one. We have performed additional analysis on this specifically in copy number balanced regions (due to the large effect of copy changes on ASE as shown in Figure 3d). Results indicate that, consistent with your expectation,

genes with nonsense mutations are indeed more likely to show ASE, and that the nonsense mutation is more likely to be on the minor expressing allele. These findings are included as Figure S8 and in the results with the following text:

“Considering specifically nonsense mutations resulting in premature stop codons (Figure S8), in balanced copy number regions these are found more frequently in genes that are ASE ($P=1.5 \times 10^{-4}$), and are more likely to be on the minor expressing allele ($P=1.6 \times 10^{-13}$); this is consistent with loss of expression of the mutated allele due to nonsense-mediated mRNA decay.”

2. In Extended Figure 6, the authors reveal cancer-specific allelic loss of gene body methylation and elevated expression on the other allele in the DUSP22 gene. It would be worthwhile to investigate whether similar relationships between alterations in gene body methylation and gene expression are observed in other genes.

We concur that this would be a very worthwhile study, and have suggested this, with the following revised text for the paragraph about gene body methylation in the discussion:

“We observed recurrent aDMRs in the intragenic promoters of RET and CDKN2A, with methylation being associated with increased expression of the canonical transcript. We further observed ASE-associated gene body methylation of DUSP22. The effect of intragenic promoter methylation on transcription is complex and bidirectional^{68–70}. Gene bodies frequently become methylated during active transcription^{68,69}, which may silence intragenic promoters via transcription interference^{68–70}. Conversely, intragenic promoter methylation can increase canonical transcript expression by reducing competition for RNA Pol binding, or through regulating transcription elongation⁶⁸, with evidence that this methylation regulates some oncogenes in cancer^{71,72}. Further analysis of intragenic promoter methylation in RET and CDKN2A is needed to determine whether this change is a mere consequence of active transcription or a key regulator of expression. A wider survey of genes showing allele-specific gene body methylation either in this cohort or others may also help to elucidate this phenomenon. Promoter methylation in the oncogene NRG1 in one case was also notable for being ecDNA-specific, a phenomenon which has not been well characterised to date⁷³.”

3. In Table 1, the KAT5 gene of POG446 is listed both in trans and in cis. Based on the context, "in cis" appears to be the correct designation.

Thank you, you are correct that the designation should be ‘in cis’ and the table has been updated.

4. Figure 3A shows that only three samples exhibit longer phased block lengths. It might be worthwhile to explore what this variation is related to (e.g. sequencing depth).

This is an interesting observation. Reviewing the sequencing coverage of these cases in Table S1, and the plotted location of these cases in Figure S7b, it appears that the strongest effect is the number of mutations (tumour mutation burden). We hypothesize this effect occurs because a very high number of somatic mutations provides additional variants which can contribute to phasing. We have added the below comment to the results:

“We found that phase block size was strongly correlated with read length (Spearman’s rho 0.72, $P \leq 2.2 \times 10^{-16}$, Figure 3a), and the cases with the longest phase blocks also had high TMB (Figure S7b).”

5. Ensuring clear correspondence between figures and Case IDs is essential. For instance, in places like Figure 2f and Supplementary Figure 1, proper annotations in figures or their legends would enhance clarity.

We fully agree! The visual table (previously Fig 2f, now Fig 2e) has been simplified, and POG identifiers added to make it more clear. We have also added POG identifiers to Supplementary Figure 1 (now Figure S3).

6. On page 12, explaining why "IDH gain-of-function mutations" cause hypermethylation would make it easier for many readers to understand. A brief explanation of the mechanisms behind this effect would improve understanding.

Response: Thank you, some additional sentences have been added to explain this; the specific IDH mutations found in the cohort have also been included in the text for clarity. The paragraph now reads:

“TET enzymes are involved in active DNA demethylation, and use α -ketoglutarate as a cofactor, which is a product of IDH enzyme activity. TET and IDH mutations are recurrent in cancer³⁷. Loss of TET demethylase activity due to loss-of-function mutations can result in hypermethylation of tumour genomes. TET can also be inhibited by accumulation of metabolites due to IDH1 R132 and other gain-of-function mutations, also resulting in tumour hypermethylation³⁷. Within the cohort, 10/189 samples (8/181 cases) had either IDH1 R132C/H or TET1/2/3 candidate inactivating mutations detected using short-read sequencing (Table S1). Compared with other cases and normal tissue, cases with these mutations show similar methylation patterns at all regulatory sites except for CGIs (Figure 4c, Figure S10). At CGIs, mutated samples show slight hypermethylation compared to the rest of tumour samples and WGBS normal samples. TET enzymes demonstrate sequence specificity toward CGIs^{37–39}. The slightly higher methylation only at CGIs in the mutated samples in our cohort supports the sequence specificity of TET enzymes and concurs with previous findings suggesting that the genome-wide hypomethylation in tumour samples is largely due to the passive DNA demethylation pathway^{40–42}.”

7. In Extended Figure 7c, the authors demonstrate significant enrichment for PRC1 and PRC2 complexes binding sites in aDMRs. Discussing the biological implications of this finding would provide valuable insights.

Thank you. We have added the following sentences to the results and discussion respectively:

“PRC1 and PRC2 are transcriptional repressive complexes involved in the regulation of developmental genes. Allele-specific methylation of their target genes in different tissues and as well as hypermethylation of those genes in cancers have been reported 47–49. This result suggests the preferential occurrence of aDMRs at genes likely involved in the stem-cell-like properties of cancer.”

“Recurrent sites of aDMRs showed an enrichment for PRC1 and 2 binding sites, congruent with their well-known role in stem cell regulation and cancer.”

8. In Figure 4g and i, ensuring consistency in abbreviations (IUM and IPU) is essential for clarity

Agreed – we have clarified these and altered for consistency in this figure, which is now Figure 5b,d.

9. In Figure 5c, 5d, and 5f, HP1(germ) and HP2(germ) appear to represent blood-derived data. It is essential to explicitly state this information in the figure legends.

That is correct, (germ) refers to blood from the same individual. This has been clarified in the figure legend.

Referees' report, second round of review

Reviewer #1:

The authors have addressed all my concerns! Great job on the revision.

Reviewer #3:

The authors responded appropriately to my comments. I believe that the quality of this manuscript has improved significantly as a result of this revision.